# Domestication reshaped the genetic basis of inbreeding depression in a maize landrace compared to its wild relative, teosinte

Luis Fernando Samayoa[1], Bode A. Olukolu[2], Chin Jian Yang[3], Qiuyue Chen[3], Markus G. Stetter[4], Alessandra M. York[3], Jose de Jesus Sanchez-Gonzalez[5], Jeffrey C. Glaubitz[6], Peter J. Bradbury[7], Maria Cinta Romay[6], Qi Sun[6], Jinliang Yang[8], Jeffrey Ross-Ibarra[9], Edward S. Buckler[7], John F. Doebley[3], James B. Holland[1,10]*

1 Department of Crop and Soil Sciences, North Carolina State University, Raleigh, North Carolina, United States of America, 2 Department of Entomology and Plant Pathology, University of Tennessee, Knoxville, Tennessee, United States of America, 3 Laboratory of Genetics, University of Wisconsin–Madison, Madison, Wisconsin, United States of America, 4 Institute for Plant Sciences and Center of Excellence on Plant Sciences, University of Cologne, Cologne, Germany, 5 Centro Universitario de Ciencias Biológicas y Agropecuarias, Universidad de Guadalajara, Zapopan, Jalisco, México, 6 Institute of Biotechnology, Cornell University, Ithaca, New York, United States of America, 7 US Department of Agriculture–Agricultural Research Service, Cornell University, Ithaca, New York, United States of America, 8 Department of Agronomy and Horticulture, University of Nebraska-Lincoln, Lincoln, Nebraska, United States of America, 9 Department of Evolution and Ecology, Center for Population Biology, and Genome Center, University of California, Davis, California, United States of America, 10 United States Department of Agriculture–Agriculture Research Service, Raleigh, North Carolina, United States of America

* Jim.Holland@usda.gov

**Data Availability Statement:** Data, analysis scripts, and supplemental table files are available at https://doi.org/10.6084/m9.figshare.14750790.

## Abstract

Inbreeding depression is the reduction in fitness and vigor resulting from mating of close relatives observed in many plant and animal species. The extent to which the genetic load of mutations contributing to inbreeding depression is due to large-effect mutations versus variants with very small individual effects is unknown and may be affected by population history. We compared the effects of outcrossing and self-fertilization on 18 traits in a landrace population of maize, which underwent a population bottleneck during domestication, and a neighboring population of its wild relative teosinte. Inbreeding depression was greater in maize than teosinte for 15 of 18 traits, congruent with the greater segregating genetic load in the maize population that we predicted from sequence data. Parental breeding values were highly consistent between outcross and selfed offspring, indicating that additive effects determine most of the genetic value even in the presence of strong inbreeding depression. We developed a novel linkage scan to identify quantitative trait loci (QTL) representing large-effect rare variants carried by only a single parent, which were more important in teosinte than maize. Teosinte also carried more putative juvenile-acting lethal variants identified by segregation distortion. These results suggest a mixture of mostly polygenic, small-effect partially recessive effects in linkage disequilibrium underlying inbreeding depression, with an additional contribution from rare larger-effect variants that was more important in teosinte but depleted in maize following the domestication bottleneck. Purging associated with the maize domestication bottleneck may have selected against some large effect

**Funding:** This work was supported by United States National Science Foundation grant IOS 1238014 to E.SB., J.F.D., J.R.I., and Q.S. (https://www.nsf.gov/index.jsp). Q.C. was supported by United States National Science Foundation grant IOS 1934865 to J.F.D. The funders had no role in study design, data collection and analysis, decision to publish, or preparation of the manuscript.

**Competing interests:** The authors have declared that no competing interests exist.

variants, but polygenic load is harder to purge and overall segregating mutational burden increased in maize compared to teosinte.

## Author summary

Inbreeding depression is the reduction in fitness and vigor resulting from mating of close relatives observed in many plant and animal species. Mating of close relatives increases the probability that an individual inherits two non-functioning mutations at the same gene, resulting in lower fitness of such matings. We do not know the extent to which inbreeding depression is due to mutations with large-effects versus small-effect polygenic variants. We compared the effects of outcrossing and self-fertilization on 18 traits in a landrace population of maize, which underwent a population bottleneck during domestication, and a neighboring population of its wild relative teosinte. Inbreeding depression was greater in maize than teosinte for 15 of 18 traits and we found that this was consistent with higher predicted 'genetic load' in maize based solely on the evolutionary conservation of the sequence variants observed in the population. We also mapped genome positions associated with inbreeding depression, identifying more and larger-effect genetic variants in teosinte than maize. These results suggest that during domestication, some of the rare large-effect variants in teosinte were bred out, but many genetic variants of small effects on inbreeding depression increased in frequency maize.

## Introduction

Darwin [1] demonstrated experimentally the detrimental effect of self-fertilization in numerous plant species, including an average reduction of 17% in the height of adult maize (*Zea mays* ssp. *mays*) plants derived from self-fertilization compared to outcrossing. The reduction in vigor and fitness of plants due to self-fertilization is an extreme example of inbreeding depression observed in many species, including humans, due to mating of close relatives. Inbreeding depression is closely related to the inverse phenomenon of hybrid vigor(heterosis), which is exploited to make high-yielding modern maize varieties [2]. Inbreeding depression typically refers to reduced vigor resulting from inbreeding within local populations whereas heterosis refers to increased vigor in crosses between differentiated populations. Thus, inbreeding depression evolves differently than heterosis in response to drift [3] and epistasis has distinct effects on the two phenomena [4]. Nevertheless, much of the genetic basis for both inbreeding depression and heterosis involves at least partially recessive deleterious variants whose effects are expressed when homozygous in inbreds or masked by dominant favorable alleles in hybrids [5–7]. Overdominance at a subset of loci contributing a small amount of variation cannot be ruled out, but repulsion-phase linkage and low recombination between recessive deleterious mutations can generate pseudo-overdominance or associative overdominance at the level of haplotypes, contributing to the retention of deleterious mutations [8,9].

A wide range of susceptibility to inbreeding depression is observed among plant species, with some species having evolved mating systems with high levels of self-fertilization accompanied by little inbreeding depression [7,10,11]. The ability of some species to tolerate high levels of self-fertilization is evidence for purging of recessive deleterious alleles during their evolution [7]. Variation for inbreeding depression also exists within species, from the sub-population to the level of individual parents [12–15]. Intraspecific variation for inbreeding depression

impacts the evolution of mating systems [16] and the response to selection in populations with mixed outcrossing and selfing [17]. The responses of genetic load and inbreeding depression to population bottlenecks are complex [18]. Some population genetic models predict that bottlenecks followed by population expansion can result in a decrease in the fixed genetic load and an increase in the segregating genetic load due to deleterious variants with small effects on fitness[18]. Other models predict nearly opposite results from bottlenecks [19],including purging of recessive large-effect variants when self-fertilization occurs [20], demonstrating the sensitivity of inbreeding depression to details of genetic architecture, mating system, and population dynamics.

Recent research has begun to identify the genomic basis underpinning inbreeding depression, revealing some large-effect recessive variants segregating in natural and breeding populations [21,22], substantial loss of transposable elements in some lineages [23], and genome-wide alterations in gene expression due to inbreeding [24,25]. Some important questions about inbreeding depression remain unresolved, however. How much inbreeding depression is due to lethal or large-effect recessive mutations vs. polygenic small effect variants distributed widely throughout the genome [6]? Although directional selection is expected to be sufficiently strong to eliminate much of the recessive genetic load from outcrossing populations, why is inbreeding depression for fitness-related traits so severe in many outcrossing species[6,7,26]? How much inbreeding depression is expressed as lethality early in development vs. reduced adult vigor and fitness [10]? Do the additional components of covariances among relatives introduced by inbreeding limit the effectiveness of selection in mixed outcrossing and selfing species like maize[27–30]?

Identifying genes underlying inbreeding depression is difficult because selection causes a negative correlation between the magnitude of variant effects and their allele frequency, limiting power of detection and estimation of their effects by genome-wide association study[31]. An alternative approach is to test the effects of groups of variants classified by population genetic or evolutionary signatures associated with deleterious effects[32]. For example, Brown and Kelly [33] demonstrated that the proportion of rare alleles (the "rare allele load") carried by inbred *Mimulus* lines was correlated to reduced fitness. In maize inbreds, rare alleles tend to have dysregulated gene expression, which is correlated with lower seed yield[34]. Variants in evolutionarily conserved sites tend to be rare in maize inbreds and complementation of such variants by common alleles in hybrids is associated with higher yield [5].

Maize was domesticated from the wild grass teosinte (*Zea mays* ssp. *parviglumis*) in southwestern Mexico about 9,000 years ago [35,36], and the extant ranges of the two subspecies overlap in Mexico and Central America [37]. Following domestication, maize experienced a population bottleneck [38,39], human selection for traits associated with easier harvest and consumption [40,41], and introgression from a weedy relative, (*Zea mays* ssp. *mexicana*) [42,43]. The shared evolutionary lineage and environmental adaptation of maize and teosinte populations that still inhabit a common range permit a comparison of the evolution of inbreeding depression in response to their distinct post-domestication selection pressures and demographies. Previously, Yang et al. [44] estimated genetic variances and correlations of 18 domestication-related traits in a teosinte and a maize landrace population sampled from nearby sites in Mexico. They demonstrated a reduction in the relative amount of additive genetic variance in maize due to the domestication bottleneck and selection, and a similar, but less extreme, reduction in dominance variance in maize across traits. Furthermore, they reported substantial differences between the trait genetic covariances (**G**-matrices) between teosinte and maize landrace and strong constraints on the direction of trait evolution imposed by the genetic correlations. Chen et al. [44] also investigated this same pair of populations,

performing a genome-wide association study to detect quantitative trait loci (QTL) using SNPs discovered from whole genome-sequencing of the parents of the population. Fewer QTL overall were discovered in maize (as expected based on that population's reduced genetic variance) but maize had more large-effect QTL, likely due to its lower effective population size and lower selection coefficients due to human management and reduced constraints from genetic correlations. Finally, most of the genetic variation for domestication-related traits in both populations appeared to be polygenic and not resolvable to individual QTL [44].

The paired teosinte and maize landrace study populations evaluated by Yang et al. [40] and Chen et al. [44] included a mix of self-fertilized and outcrossed progenies, providing an opportunity to study the differences in inbreeding depression between these two populations. Maize and teosinte are both predominantly outcrossing species that also naturally self-fertilize at a low rate [45,46]. Modern maize hybrids are created by first selecting highly homozygous inbred lines, followed by crossing unrelated lines to restore heterozygosity across much of the genome, resulting in hybrid vigor [2]. Selection during inbreeding and among resulting inbred lines provides ample opportunity for purging large effect deleterious variants as well as polygenic genetic load, since commercial inbreds are selected for relatively high levels of pollen or seed production [47]. The rate of genetic gain for yield in maize hybrids was greater than in maize inbreds, resulting in an increase in the amount of inbreeding depression over generations of yield selection in modern maize [48,49]. In contrast to commercial hybrids, maize landraces are maintained as open-pollinated populations selected for local adaptation as well as for ear and kernel type, although migration among populations arising from seed exchange among farmers is also common [50–53]. Similarly, teosinte populations are predominantly outcrossing; however, reductions in teosinte habitat area may result in inbreeding due to small effective population sizes [37,54].

The different demographic histories of maize landraces and teosinte are expected to affect their genetic load of deleterious alleles. Wang et al. [55] and Lozano et al. [56] reported that maize has higher predicted deleterious genomic burden than teosinte, estimated based on evolutionary conservation [57] or the predicted effects of amino acid substitutions [58] of SNPs. Importantly, Wang et al. [55] demonstrated that maize has more fixed putative deleterious sites than teosinte, but such fixed deleterious variants reduce the fitness of both outbred and inbred progenies equally, thus they do not contribute to inbreeding depression. In contrast, maize has fewer segregating predicted deleterious sites than teosinte [55], leading to the expectation that maize should exhibit less inbreeding depression than teosinte [59]. Evidence for the functional importance of this predicted segregating load in maize is that hybrid vigor in crosses between elite maize inbred lines can be largely attributed to complementation of the predicted SNP burdens of different inbred parents [5].

In this study we investigated the genetic architecture of inbreeding depression in the same paired populations of maize and teosinte previously studied by Yang et al. [40] and Chen et al. [44]. The plants evaluated in the field were derived by self- and cross-mating samples of 40 and 49 outbred plants sampled from the maize landrace and teosinte populations, respectively, permitting estimation of the effects of inbreeding on the trait means and genetic variances. Yang et al. [40] previously used marker-based realized additive and dominance genomic relationships among the individuals within each population to estimate genetic variance and covariance estimates. Such estimates are more accurate than pedigree-based estimates [60], but the models require some simplifying assumptions that ignore the effects of inbreeding. Here we applied an alternative pedigree-based quantitative genetic model that partitions the genetic variance separately for outbred and inbred progenies to reflect the potential influence of additional components of genetic covariances that come into play under inbreeding [61]. The pedigree-based diallel model facilitates estimating the effect of self-fertilization on trait

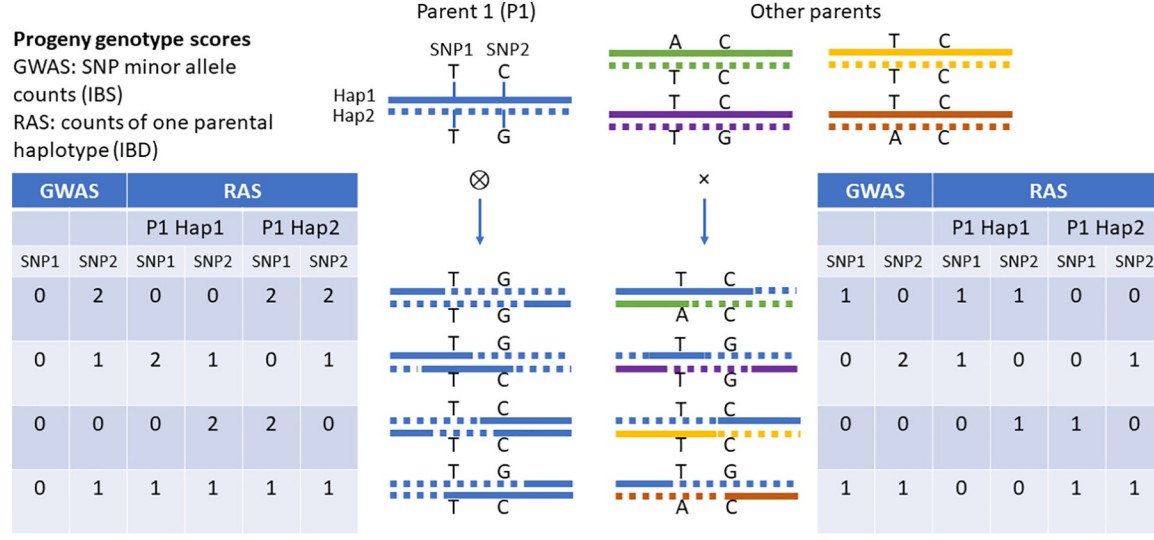

**Fig 1. Diagram of key features of rare allele linkage scan compared to genome wide association study tests for quantitative trait loci.**

means and parental breeding values. Strong positive correlations of parental breeding values between outcross and selfed progenies are expected when genetic variation in both outcrossed and partly inbred progenies is due mostly to additive genetic variance, which can occur even if dominance effects are sufficiently strong to cause substantial inbreeding depression. Further, the pedigree-based model employed here permits testing if the micro-environmental variances for outbred and inbred progenies are equal, which may not be true if inbreeding reduces homeostasis[62], and testing for the importance of reciprocal genetic effects that can arise due to maternal, cytoplasmic, or epigenetic effects [63].

We used SNPs discovered from sequencing the parents of the two populations to predict segregating genetic load based on evolutionary conservation of SNP variants. Wang et al. [55] previously reported that maize has more segregating deleterious variants than teosinte, but they sampled different individuals than the parents of our populations. Here we estimate fixed and segregating predicted load directly for our population parents and compare that to the observed phenotypic inbreeding depression. We developed a novel QTL mapping strategy ("rare allele scan"; Fig 1), aimed at detecting the effects of alleles private to individual parents, as deleterious alleles contributing to inbreeding depression are expected to be rare. The large family sizes in these populations provide good power for estimating phenotypic effects of variants carried by individual parents, even if the variants are rare in the larger populations from which the parents were sampled. This analysis complements the previous standard GWAS performed by Chen et al. [44], which is optimized for detecting variants with moderate allele frequency that influence overall phenotypic variation, but has limited power to detect loci where a causal variant is carried by only a single parent. We also identified putative juvenile-stage lethal alleles based on linked segregation distortion and estimated the phenotypic effects of parental rare allele burden in the maize and teosinte populations to understand how domestication and the different histories of these populations influenced inbreeding depression and its genetic architecture.

## Results and discussion

### Maize exhibits more inbreeding depression than teosinte

We evaluated 4,455 teosinte plants grouped into 49 selfed and 377 outcross full-sib families, and 4,398 maize landrace plants grouped into 34 selfed and 89 outcross full-sib families for 18 traits related to domestication (S1 and S2 Tables)[40]. The 18 measured traits were grouped into vegetative/flowering time, environmental response, and reproductive categories (Table 1) [40]. The selfed and outcrossed progeny means were estimated while accounting for parental relationships and non-genetic effects using a diallel (parental pedigree-based) model. The coefficient of inbreeding depression was measured as the absolute proportional change in population mean from a single generation of selfing, and we ignored tiller number (TILN) in summaries of inbreeding depression by trait group, as its very low mean value in outcross progenies in maize greatly inflates the proportional change. The coefficient of inbreeding was higher on average for environmental response traits within teosinte (19%) and for reproductive traits in maize (32%; Figs 2; S1 and S2; S3 Table). Maize had greater inbreeding depression than teosinte for 15 of 18 traits, for each trait category mean, and had about twice as much inbreeding depression for reproductive traits (S3 Table). Inbreeding depression and dominance variance are influenced by dominance effects in distinct ways [4], reflected in the strong but imperfect correlation across traits ($r = 0.63$ in teosinte, $r = 0.68$ in maize; S1 Fig) between the coefficient of inbreeding and the proportion of genetic variance due to dominance previously estimated by Yang et al. [40].

The higher inbreeding depression in maize is initially surprising given the lower number of segregating deleterious variants in maize predicted on the basis of evolutionary conservation by Wang et al.[55]. To investigate patterns of deleterious variation predicted from sequence conservation alone, we used Genomic Evolutionary Rate Profiling (GERP) scores calculated by Kistler et al.[64] to identify putatively deleterious SNPs in our populations. Congruent with Wang et al. [55] and with the higher diversity within teosinte, we identified a total of 362,145

**Table 1. Trait Abbreviations.** List of 18 teosinte-maize landrace comparable traits and the corresponding acronyms, units and trait groups. From Yang et al. (2019).

| Trait | Acronym | Units | Trait Group |
|---|---|---|---|
| Days to Anthesis | DTA | days | Veg/FT |
| Days to Silking | DTS | days | Veg/FT |
| Plant Height | PLHT | cm | Veg/FT |
| Leaf Length | LFLN | cm | Veg/FT |
| Leaf Width | LFWD | cm | Veg/FT |
| Tiller Number | TILN | count | EnvRes |
| Prolificacy | PROL | count | EnvRes |
| Lateral Branch Node Number | LBNN | count | EnvRes |
| Lateral Branch Length | LBLN | mm | EnvRes |
| Lateral Branch Internode Length | LBIL | mm | EnvRes |
| Ear Length | EL | mm | Rep |
| Cupules per Row | CUPR | count | Rep |
| Ear Diameter | ED | mm | Rep |
| Grains Per Ear | GE | count | Rep |
| Ear Internode Length | EILN | mm | Rep |
| Total Grain per Plant | TGPP | count | Rep |
| Total Grain Weight per Plant | TGWP | g | Rep |
| Grain Weight | GW | mg | Rep |

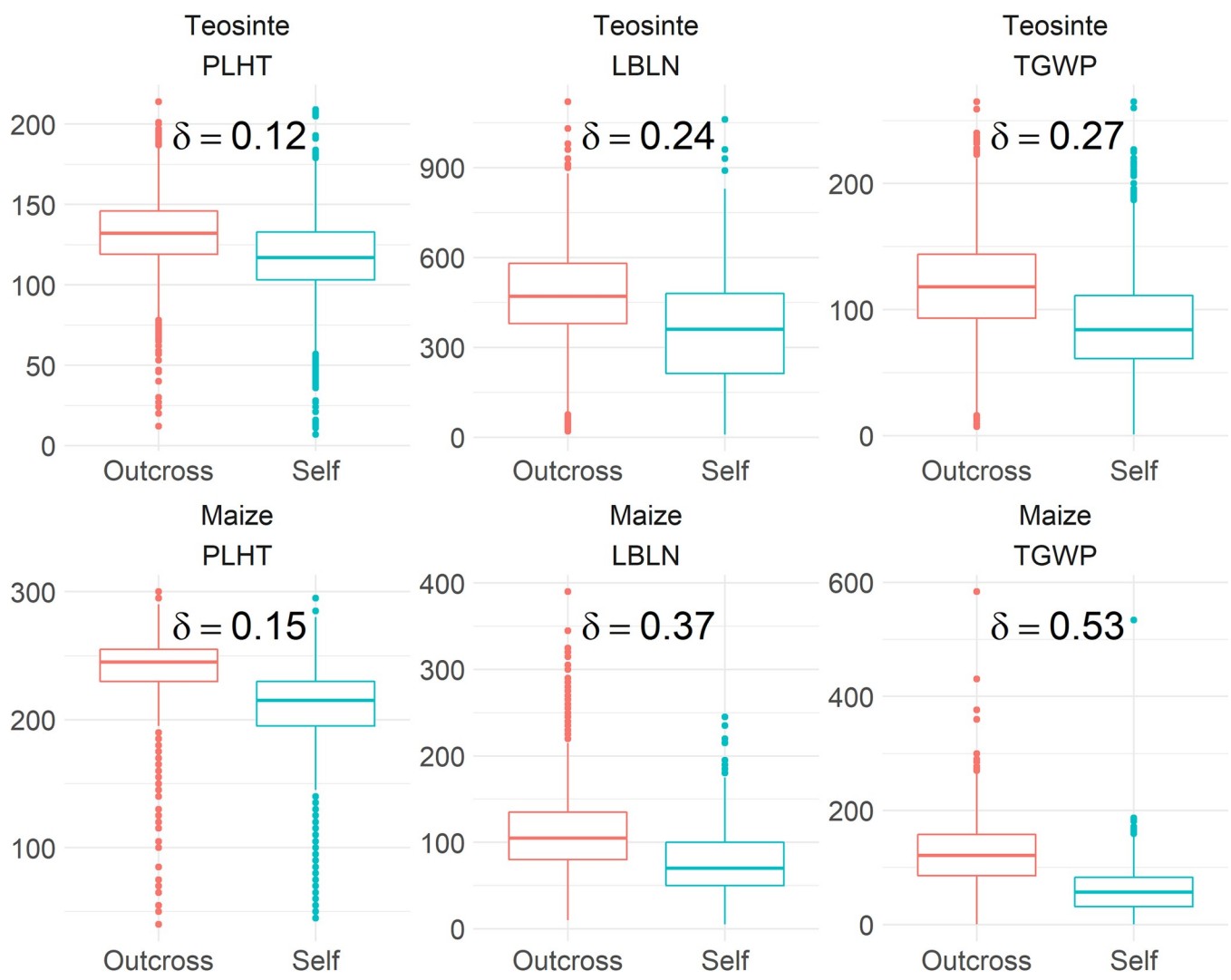

**Fig 2. Distribution of representative traits plant height (PLHT, a Vegetative/Flowering Time trait), lateral branch length (LBLN, an Environmental Response trait), and total grain weight per plant (TGWP, a Reproductive trait) among outcross and selfed progenies in teosinte and maize.** The coefficient of inbreeding (δ) represents inbreeding depression as the mean proportional decline in trait values due to one generation of self-fertilization.

vs. 293,720 variants with GERP scores segregating within teosinte and maize, respectively. The number of segregating deleterious sites is not the sole determinant of inbreeding depression, however, because mutations vary for both their predicted effects on fitness and their allele frequency. Whereas the total number of deleterious SNPs in the population is higher in teosinte, the number of sites at which an individual parent carries a predicted deleterious variant is higher in maize (Figs 3A and S3; S4 Table), indicating that the frequencies of deleterious variants is higher in maize. Considering the predicted magnitude and zygosity of each variant, the mean burden of each of these sites is slightly lower in maize (Fig 3B). The total predicted mutational burden per parent, which depends on both the number and effect of deleterious alleles carried by each parent, is higher in maize than in teosinte (Figs 3C and S3 and S4 Table), congruent with the generally greater inbreeding depression observed in maize. Taken together, these observations suggest that individual maize plants tend to carry a larger number of more

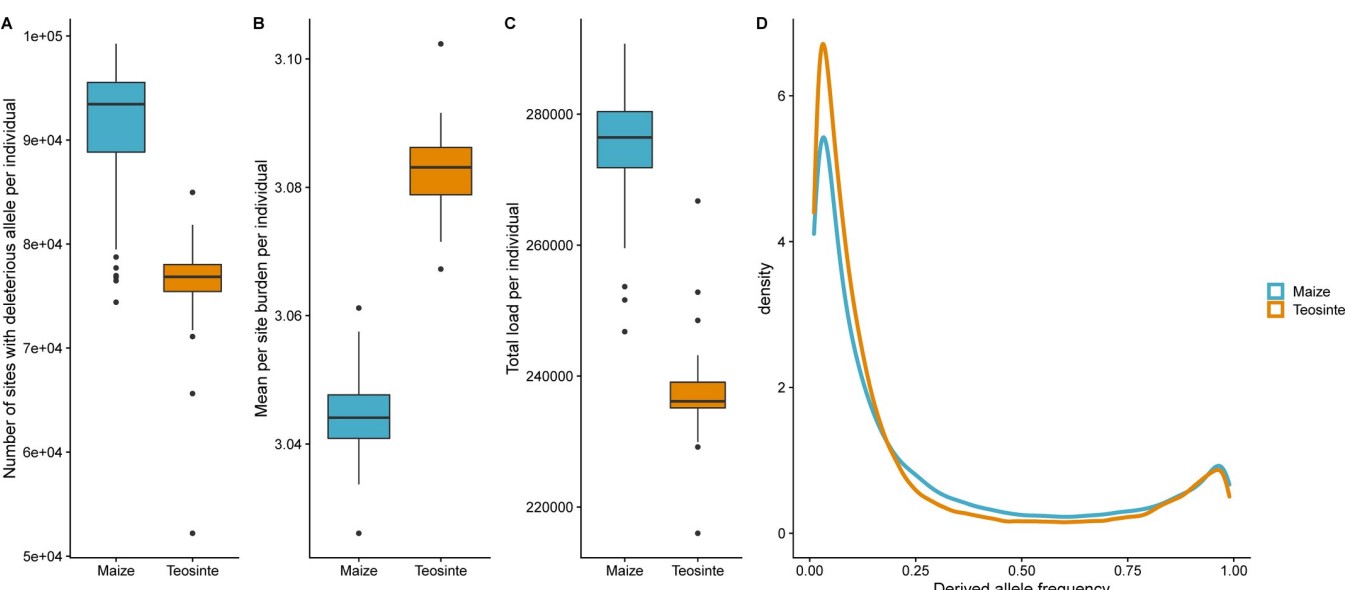

**Fig 3. Predicted segregating mutational burden per parent in the parents of maize and teosinte populations.** (A) Total number of sites heterozygous or homozygous for segregating deleterious alleles per individual parent, (B) Mean burden per site per parent based on genomic evolutionary rate profiling (GERP) score under a model of partial recessivity, including only sites segregating within the parent's population, (C) Total burden per parent based on GERP scores, (D) Derived allele frequency spectrum for maize and teosinte populations. A total of 293,720 and 362,145 variants with GERP scores are segregating within maize and teosinte, respectively.

weakly deleterious variants than their teosinte counterparts. Consistent with this, the average frequency of derived alleles is higher in maize compared to teosinte (Fig 3D).

Why is the mean segregating genetic burden greater in this maize population than its closest wild relative population? Our results suggest that although maize has fewer deleterious variants than teosinte (due to an overall reduction in diversity following the domestication bottleneck), they occur at higher frequency in maize than in teosinte, increasing the mean segregating burden. Simons et al. [18] found that population bottlenecks followed by growth are expected to decrease the fixed burden and increase the segregating burden due to weak recessive deleterious alleles, suggesting that our observed results can largely be explained by drift resulting from the domestication bottleneck. In addition, pseudo-overdominance generated by repulsion phase linkages among deleterious variants appears to be widespread in maize [5,65–67] and in particular, genomic regions with low recombination rates can shelter deleterious variants dispersed in repulsion phase [68], permitting a buildup of genetic load that may have initially accumulated through drift [59]. In addition, Chen et al. [44] suggested that selection coefficients for many traits in maize could be lower than in teosinte because human selection is often aimed at traits other than the domestication-related traits measured in this study, and because the structure of trait genetic correlations in maize results in generally lower correlations of most traits with fitness.

The 15% decrease in height and the increase in days to flowering resulting from selfing in the maize landrace is very similar to that reported by Hallauer and Sears[69] and Cornelius and Dudley[70] using open-pollinated populations in the North Central corn belt region of the USA. The more than 50% decrease in total grain weight per plant (TGWP) due to selfing observed here is greater than the about 40% decrease reported in the earlier studies, however. The population studied by Hallauer and Sears [69] was founded by inbred lines, thus excluding the possibility of segregating recessive lethal alleles. It is likely that landrace populations of

maize are segregating for more deleterious variants than are breeding populations that have been selected for productivity using randomized and replicated family-based breeding evaluations. Landrace maize is often selected for very specific ear and kernel forms and culinary uses [41,52], and this may also reduce selection on traits related to productivity and yield in landraces.

## Linear and non-linear responses to inbreeding

We also tested the linear and quadratic regressions of individuals' phenotypes on their genomic inbreeding coefficient ($\beta_F$) estimated from markers, while accounting for relatedness among the individuals with genomic relationship matrices [40]. This method uses the fine-scale variation in realized inbreeding values among progenies (S4 Fig) and the quadratic regression term serves as a test for epistasis underlying inbreeding depression [4,62]. The linear regression coefficients were almost exactly twice the estimated coefficients of inbreeding (S1 Fig; S3 Table) because the coefficient of inbreeding was calculated from one generation of selfing that causes inbreeding to an average of 50% of complete homozygosity. The quadratic regression coefficient was significant ($p < 0.05$) for plant height (PLHT), leaf width (LFWD), lateral branch length (LBLN), and lateral branch internode length (LBIL) in teosinte but not for any trait in maize. We infer that epistasis was not an important component of the genetic architecture of inbreeding depression for most traits in teosinte and all traits in maize. For the four traits where epistasis appears to be an important component of inbreeding depression, the regression slopes for the quadratic terms were all negative (in the same direction as the linear term), indicating that inbreeding depression accelerated with increasing homozygosity. This is evidence of synergistic effects among deleterious mutations, which may arise due to genetic complementation[71,72]. For example, a deleterious mutation at one locus may be masked by functional alleles at complementary loci, so that genomic duplication buffers against the early generations of inbreeding, but this buffering can break down as the probability of simultaneous homozygosity at complementarily interacting loci increases at higher levels of inbreeding. The loss of epistasis as a component of inbreeding depression for PLHT, LFWD, LBLN, and LBIL in maize could be the result of the domestication bottleneck. Yang et al.[40] previously demonstrated a widespread reduction in additive genetic variance when comparing the maize and teosinte populations studied here. The reduced effective population size associated with the domestication bottleneck is expected to decrease epistatic variance even more strongly than additive variance[61].

## Reciprocal genetic effects and changes in residual variances associated with inbreeding

The diallel model employed in this study allows estimation of quantitative genetic parameters beyond the coefficient of inbreeding, including reciprocal genetic variances, separate estimates of the micro-environmental residual variances for the selfed and outcrossed progenies, and correlations between parental breeding values in the selfed and outcrossed progenies. Separate estimation of the residual non-genetic variation associated with selfed vs. outcrossed progenies permits a test of the hypothesis that outbred and inbred plants differ for environmental variance [62,73]. Our results demonstrate that environmental variance increased with inbreeding on average for maize vegetative and reproductive traits (mean increases of 40% and 20%, respectively), suggesting that partially inbred maize plants have reduced capacity for homeostasis for these traits. In contrast, residual environmental variance decreased on average with inbreeding for maize environmental response traits (mean decrease of 39%) and all groups of

teosinte traits (decreased from 3% to 29%); these reductions may occur because of reduced scale associated with inbreeding.

Reciprocal genetic effects can include cytoplasmic, maternal, and epigenetic parent-of-origin effects [63], which have been proposed to impact the trajectory of evolution [74,75], including in plants [76]. Based on model fit criteria, we observed no evidence for reciprocal genetic variance for any trait in either population. This suggests that the potential for maternal and parent-of-origin effects to shape the response to selection within these populations is limited.

A feature of our design is the ability to estimate the breeding values for each parent in both their outcross and inbred progenies (S5 Table) and their covariance. We expect the correlation of a parent's breeding value between outcrossed and selfed progenies to equal 1 if all the genetic variance is additive and less than 1 when dominance and other higher order covariances are important. The correlation between parental breeding values across outbred and inbred progenies was $r > 0.90$ for all traits except EL in maize and LBIL in teosinte (Fig 4). These results reflect the fact that even though dominance effects are important for many of the traits measured (demonstrated by the significant inbreeding depression), most of the genetic variance is additive, which is true across a wide range of genetic architectures[77]. The diallel model also permits estimation of additional components of genetic covariance that occur under inbreeding: the covariance of additive and dominance effects in inbred individuals ($\sigma_{ADI}$) and the variance of dominance effects in inbred individuals ($\sigma_{DI}^2$) [30,61], but the estimates had poor precision (S6 Table), preventing reliable inferences. A method to use population sequence data to estimate higher-order identity-by-descent coefficients has been developed [78] and could be used to more directly estimate higher-order covariance components, but this approach is very computationally intensive and will be investigated in future efforts.

The very high correlations between outbred and inbred progeny breeding values measured here contrast with low to moderate correlations estimated for grain yield, plant height, and grain weight from breeding populations by Cornelius and Dudley [70] and Coors [79]. These comparisons, along with much larger proportions of dominance variance estimated for yield in several open-pollinated maize breeding populations [28,80] suggest that intensive selection and coincident bottlenecking due to maize breeding in those populations depleted additive genetic variation, leading to relatively higher proportions of dominance variance while decreasing inbreeding depression at the same time. Thus, it appears that some important components of the genetic architecture of inbreeding depression in maize differ between landrace and modern breeding populations, due to their different histories of population management and selection.

## Rare allele effects detected by linkage scan

The rare allele linkage scan is optimized to detect the effect of a private allele carried by a single parent with a contrasting effect to the common alleles carried by other parents. Thus, this linkage scan estimates local effects of each haplotype carried by each parent (Figs 1 and 5, S5–S10; S7 Table). More than twice as many rare allele QTL were detected in teosinte (468) than maize (173; S11–14 Figs, S7 Table). This result is in line with that of Chen et al. [44] who discovered many more QTL in teosinte than maize via a GWAS geared at detecting more commonly segregating variants. The relative proportion of QTL detected for different trait groups was nearly identical within maize (29–36% of QTL in each group), but within teosinte, most QTL were detected for reproductive traits (58%, compared to 16% for environmental response and 26% for vegetative/flowering time traits; S14 Fig).

## Inbred vs Outcross Family Means

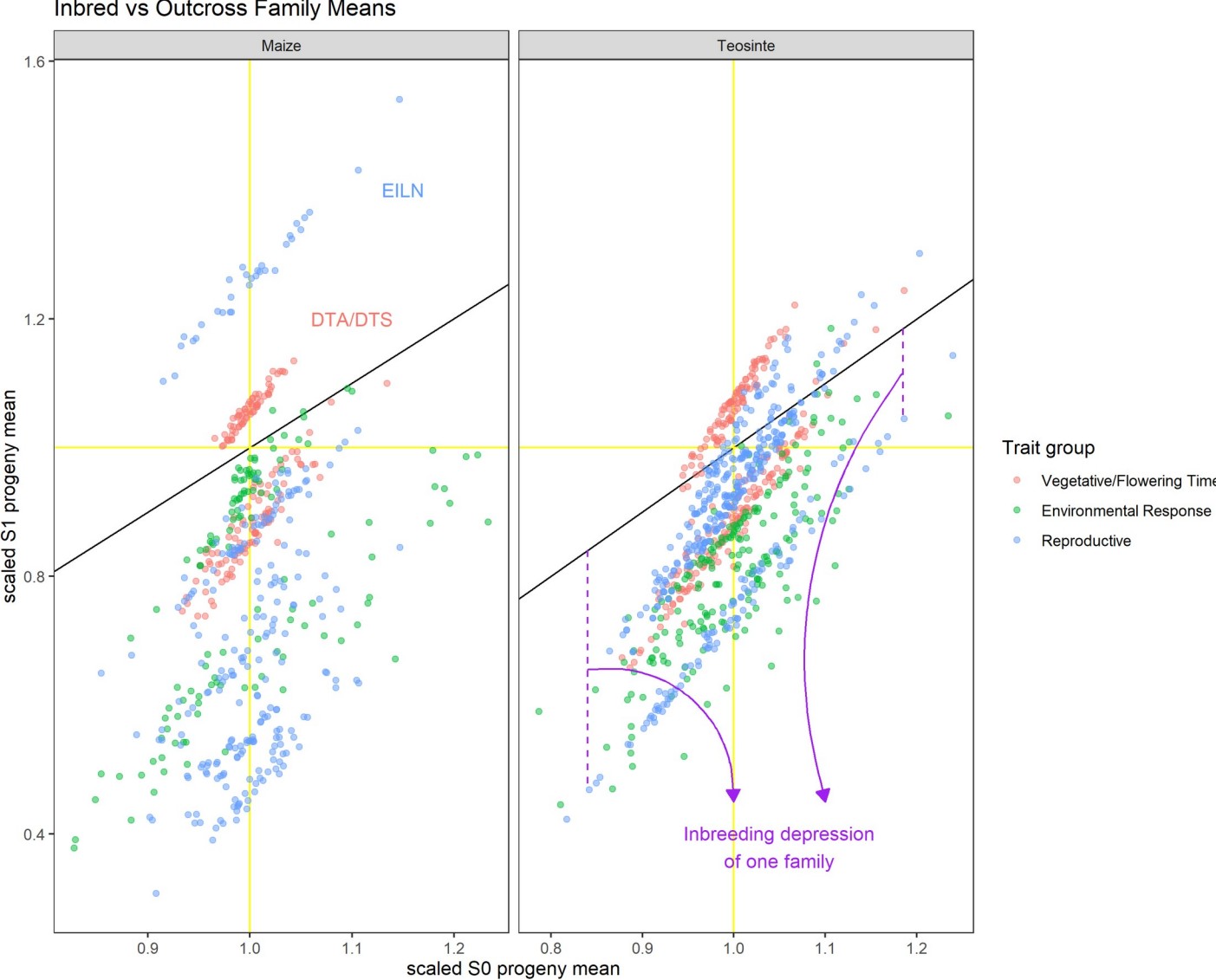

**Fig 4. Variation in outbred and selfed progeny breeding values among parents in teosinte and maize.** Mean values for self-fertilized S1 progenies (Y-axis) and outbred progenies (X-axis) scaled to the outbred population mean for each parent. Each trait is plotted in a different color. The 1:1 line is plotted in black, deviations from this line in the vertical direction correspond to the inbreeding depression for a single parent's selfed progenies compared to their outcrossed siblings.

Against the overall trend of more QTL segregating in teosinte, Chen et al. [44] identified 7 and 12 large-effect QTL (defined as having > 1 phenotypic standard deviation effect) in teosinte and maize, respectively, suggesting that reduced selection coefficients in maize permit functional variants to be maintained at higher frequencies by drift. In the current study, we identified many more large-effect QTL that are by definition rare, and therefore are harder to detect in a standard GWAS. In contrast to the previous GWAS results, teosinte had more large-effect rare allele scan (RAS) QTL (58) than maize (49; S14 Fig). The primary difference was observed for reproductive traits where teosinte had 28 large-effect RAS QTL compared to 18 for maize (S14 Fig). These QTL may correspond to variants that are maintained only at very low frequency due to stronger selection on reproductive traits; teosinte has a higher proportion of rarest class of alleles and by chance some very rare alleles were sampled among the study parents.

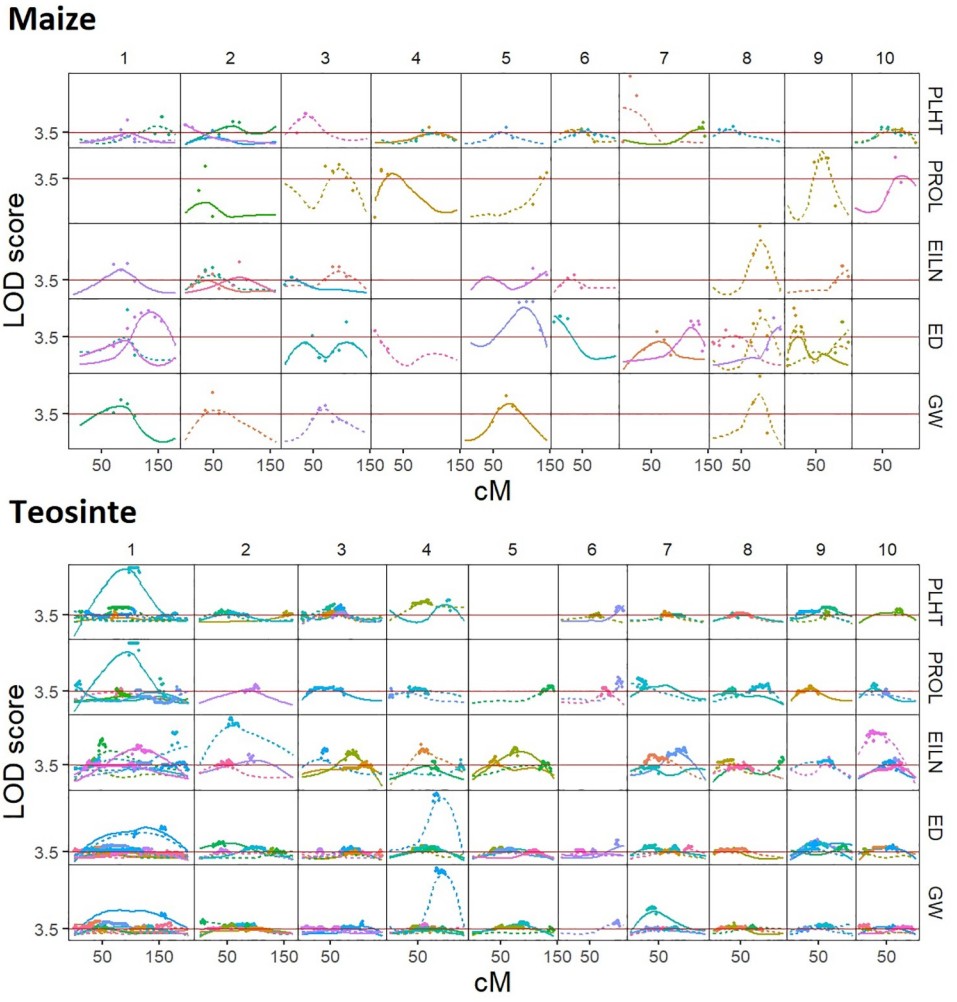

**Fig 5. Rare allele linkage scan for five traits in maize landrace and teosinte populations.** Each column represents one chromosome; linkage map cM positions are displayed on the x-axis. Each row represents LOD scores of linkage scan for one trait. Each founder parent is represented by a different color and the two different haplotypes of each parent are distinguished by solid vs. dashed lines. Red solid line indicates the whole-genome LOD significance threshold.

Numerous QTL affecting multiple traits were detected (S8 and S9 Tables), reflecting pleiotropic or linked QTL. For example, a QTL allele on chromosome 8 carried by maize parent 164_8 affected six reproductive traits: cupules per row (CUPR), ear diameter (ED), grains per ear (GE), ear internode length (EILN), total number of grains per plant (TGPP), and weight per grain (GW; Fig 5); the rare founder allele reduced the ear diameter and number of seeds, while increasing EILN and weight of the fewer kernels produced per plant (GW; S8 Table). The plants inheriting the rare allele from teosinte founder PC_O51_ID2 at a QTL on chromosome 6 were shorter and later-flowering with smaller ears (S9 Table).

The mean size of 2-LOD support interval across all QTL was 38 cM for maize and 26 cM for teosinte, indicating generally low resolution of the parent-specific rare allele QTL, due to the reliance of the RAS on within-family linkage. There was no evidence for QTL tending to occur in low-recombination regions, but the low resolution of QTL positions precludes making strong inferences from this result.

To check how the sample size of progenies (which varies among parents) influences the rare allele scan results, we estimated the correlation between the number of QTL detected and the number of progenies per parent, observing moderate correlations within each population ($r = 0.38$, $p = 0.04$ in maize, $r = 0.45$, $p = 8.3$ x10-04;S15 Fig). Nevertheless, more total QTL and more QTL per parent (10 QTL per parent in teosinte vs 5 per parent in maize on average) were detected in teosinte despite fewer offspring per parent (231 on average in maize compared to 182 on average in teosinte). Therefore, differences in family size between maize and teosinte do not appear to account for the larger number of QTL detected in teosinte.

Relatively few traits exhibited evidence for shared QTL positions between maize and teosinte, suggesting that most private allele variants detected with the RAS were unique to each population (S16 Fig). However, we found significant non-random overlaps between RAS QTL intervals and individual SNPs previously detected by Chen et al. [44] with GWAS for most traits in teosinte and some traits in maize (S5 - S10 and S16 Figs; S10–S12 Tables). Over all traits, 48% of maize and 69% of teosinte RAS QTL encompassed GWAS QTL (S12 Table). This suggests that the two methods identify distinct but overlapping parts of the genetic architecture. Further evidence of this are the significant correlations between the standardized effects of rare allele QTL and minor alleles detected by GWAS within common QTL intervals ($r = 0.49$ and $0.60$ for additive and dominance effects in maize; $r = 0.32$ and $0.57$ for additive and dominance effects in teosinte, all $p < 0.0001$; S17 Fig). In particular, Chen et al. [44] demonstrated that most large-effect QTL had low minor allele frequency, presumably due to selection against large effect variants; the largest of those large-effect SNP associations were also detected in our RAS as QTL, for example the major plant height QTL on chromosome 7 in maize and the major grain weight QTL on chromosome 4 in teosinte (Fig 5).

The additive and dominance effects of RAS QTL were also strongly correlated ($r = 0.83$ and $0.73$ in maize and teosinte, respectively, $p < 0.0001$; S18 Fig), congruent with GWAS results [44]. Thus, the preponderance of QTL effects detected either by GWAS or by RAS had dominance effects in the same direction as the major allele and were consistent with the direction of observed inbreeding depression for the trait (S19 Fig). In contrast to the general trends of positive major allele effects on reproductive traits, we nevertheless observed one GWAS association each for TGWP in maize and GW in teosinte and two RAS QTL for ED in teosinte with standardized major allele additive effects less than -1.5, representing rare favorable alleles for these traits; the dominance effects in all these cases were also negative, indicating that the rare favorable alleles were also recessive. These QTL cause effects counter to genome-wide trends for inbreeding depression.

## Rare QTL allele effects are associated with a minority of genetic variation and inbreeding depression

Although we detected many rare allele QTL effects, the total proportion of trait variation that can be attributed to these QTL is low for all traits (Fig 6). Most of the variation is due to polygenic background effects (modeled here using principal components of the marker data), with rare-allele scan QTL contributing an average of 26% (range 9% to 54%) of the total genetic variance in maize and 30% (range 23% to 41%) in teosinte. Within both populations, the mean proportion of genetic variance due to RAS QTL was lowest for reproductive traits (24% to 25%) and largest for environmental response traits (33% in maize and 35% in teosinte). This may reflect stronger selection pressure against large effect variants affecting reproductive traits due to their higher correlation with fitness. Rare QTL effects were validated by measuring their out-of-sample prediction ability using cross-validation (S13 Table). The correlation coefficient between predicted and observed values in the validation set increased on average by

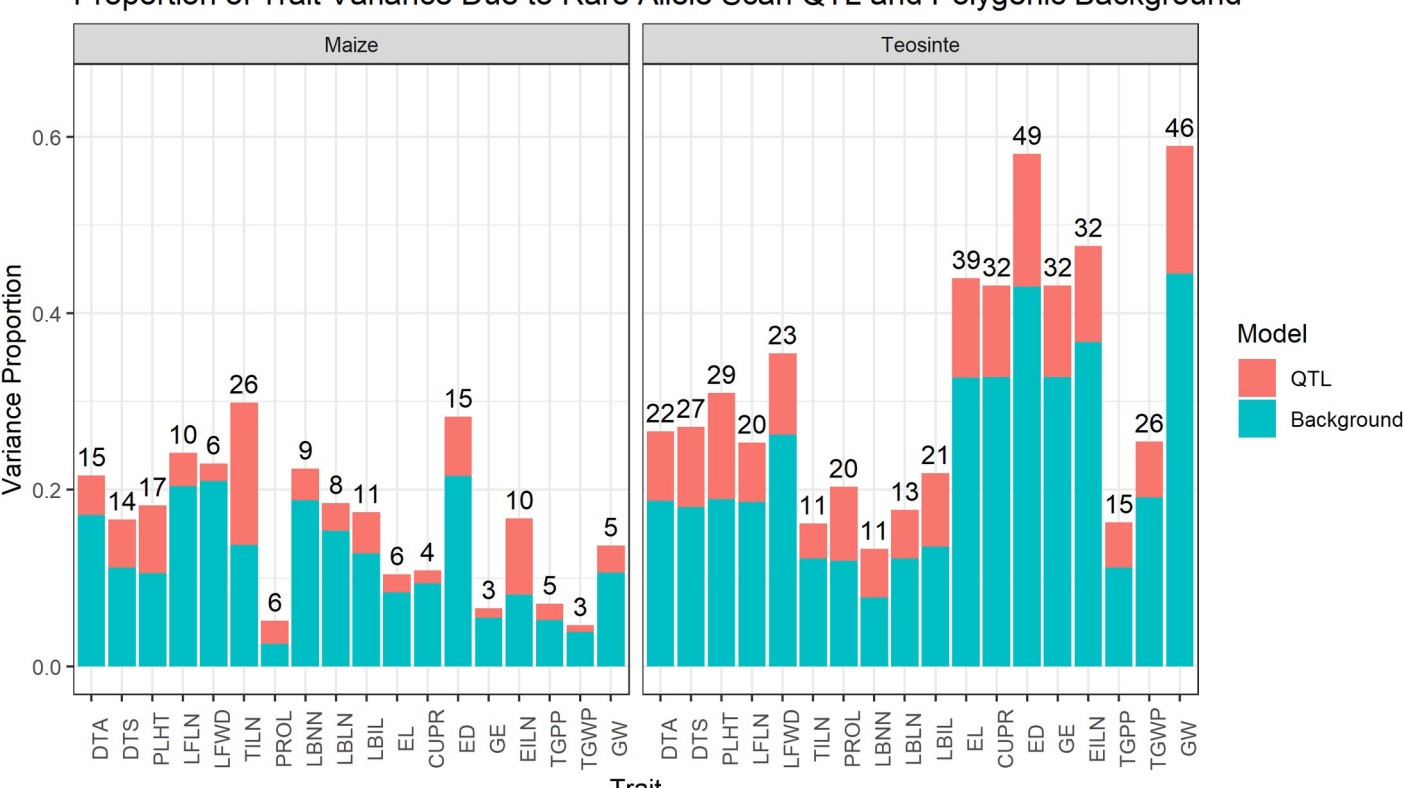

**Fig 6. The proportion of trait variance due to rare allele scan QTL (red color) and polygenic background effects (blue color) for each of 18 traits measured in maize and teosinte populations.** The number of rare allele scan QTL detected for each population and trait is indicated on top of bars.

0.13 (teosinte) and 0.04 (maize) when adding RAS QTL to a model with no genetic effects increased and by 0.03 (teosinte) and 0.01 (maize) on average when adding rare allele scan QTL to a model with principal components (S13 Table). The drop in prediction improvement when QTL are added to a model with polygenic effects demonstrates the shared variation that can be accounted for either by principal components or QTL.

In general, the amount of observed inbreeding depression that could be accounted for by RAS QTL was below 20%, and the predicted direction of inbreeding depression was wrong in a handful of cases (S20 Fig). The best predictions were made for EILN in teosinte and TILN in maize, for which between 40 and 50% of the observed inbreeding depression was predicted by the RAS QTL. GWAS associations predicted a bit more of the observed inbreeding depression, but again this was a relatively small proportion of the total inbreeding depression for most traits. In general, the predicted inbreeding depression was greater for teosinte than maize, probably due mostly to the much larger number of QTL detected in teosinte. In summary, our results indicate that we can detect rare variants contributing to inbreeding depression, but most variation for inbreeding depression appears to be due to polygenic small effect variants that are difficult to detect with either GWAS or RAS, are not easily purged by natural selection, and have increased in importance in maize compared to teosinte.

## Early seedling lethal variants inferred from segregation distortion

We sampled leaf tissue about one month after sowing; during this time 15% of maize plants and 49% of teosinte plants did not germinate or survive the first month to be sampled for leaf

tissue. Of the plants that survived the first month, an additional 10% of maize plants and 3% of teosinte plants died before maturity. Strongly deleterious alleles, therefore, may have been purged from our self-fertilized families before they could contribute to inbreeding depression of the traits we measured on adult plants. We should be able to detect such loci by testing for segregation distortion in the selfed progeny of each parent separately. We caution, however, that although we removed the seed coat, teosinte varies for seed dormancy, a trait adaptive in natural populations[81], and it is possible that some seeds that did not germinate may have been alive but dormant. We identified 3 significant segregation distortion regions (SDR) across three different chromosomes in 3 self-families in maize landraces and 16 significant SDR on seven different chromosomes in 11 self-families in teosinte (Fig 7; S14 Table). We checked for correspondence between SDR and six QTL for germination rate in teosinte previously identified by Chen et al. [44]: three of the six QTL mapped to the same chromosomes as SDR, one of them inside an SDR (chromosome 1) and one near an SDR (chromosome 5; S21 Fig). These last two cases may represent SDR caused by seed dormancy rather than by selection against lethal recessives, leaving 14 SDR not closely linked to a seed dormancy QTL.

Segregation distortion was mostly observed against either one or two homozygous classes in both populations (S22 Fig), consistent with the expectation that lethal or strongly deleterious alleles are recessive and maintained only in heterozygotes. Some SDRs exhibited a deficit of both homozygous classes, consistent with multiple deleterious recessive variants carried in repulsion phase linkage in the parent of the affected family. An exception to this expected pattern was observed in five of the 16 SDRs identified in teosinte, where the heterozygote class was deficient (S22 Fig). Some of these cases could represent deleterious alleles that are not recessive (such as the SDR allele on chromosome 7 carried by parent PC_I05_ID1 which is completely absent in its selfed progenies). We also noted that three of the SDR detected in teosinte corresponded with the three best characterized gametophyte factor loci (*Ga1*, *Ga2*, and *Tcb1*; S23 Fig) known to control preferential fertilization [82] in teosinte [83,84].These loci may contribute to reproductive isolation between populations and could be polymorphic in some populations due to complex balancing selection [83,85]. Self-fertilization of individuals heterozygous at these loci is expected to result in reduced frequency of one allele due to selection against pollen carrying one of the two alleles, with no selection occurring on the female gametes. The observed patterns of segregation distortion at these loci did not match this pattern (S23 Fig), however, suggesting that they are not the direct result of gametophyte factors, but instead due to accumulation of deleterious alleles linked to 'selfish' gametophyte alleles.

All but five of the 19 SDRs across both populations were detected in regions with lower recombination rates than their respective chromosome average (S14 Table and S24 Fig). The observed correspondence between low recombination rates and regions carrying strongly deleterious alleles that are eliminated before the adult life stage is consistent with the prediction that selection is less effective in lower recombination regions [59,86] and previous observations in maize inbred lines [68]. This result contrasts with the lack of a relationship between recombination rate and RAS QTL, perhaps because the individual QTL effects on fitness are smaller than the SDR effects, and these QTL are under weaker selection.

## Parental rare allele load is not consistently related to inbreeding depression

We compared the genome-wide rare allele load of each parent to its percent inbreeding depression for each trait, measured based on its selfed progenies. No relationship was observed for most traits, but significant ($P < 0.05$) positive correlations between the proportion of loci carrying rare alleles in a parent and the inbreeding depression measured in its offspring was observed for days to silk (DTS) and GW in maize ($r = 0.37$ and $0.41$, respectively; Fig 8;

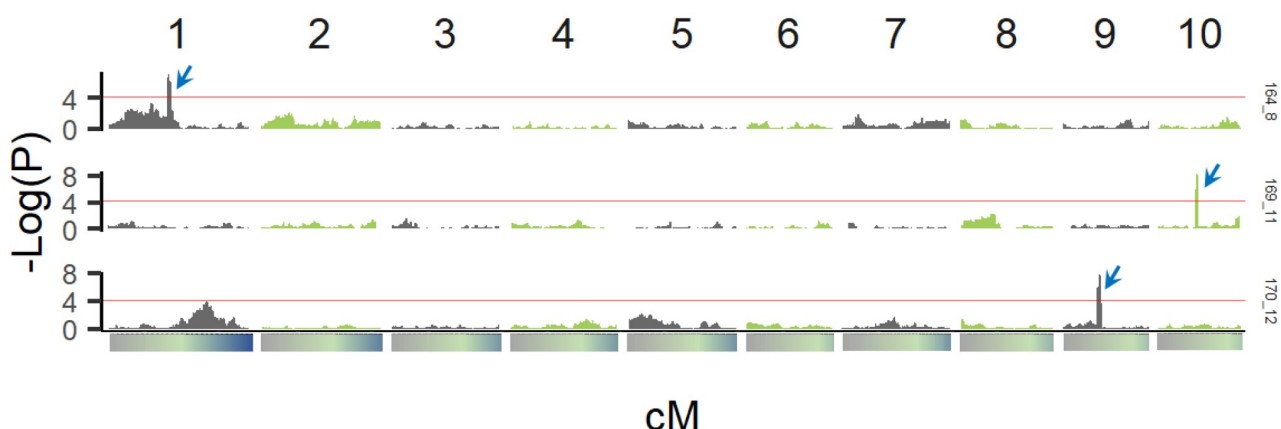

**A. Segregation Distortion Regions in 3 Maize Self-families**

**B. Segregation Distortion Regions in 11 Teosinte Self Families**

**Fig 7.** Segregation distortion whole genome scan results plotted as -log$_{10}$p-valueof the chi-square test for agreement to expected 1:2:1 segregation at each genome region (cM position) for (A) three selfed families in maize, and (B) eleven selfed families in teosinte; each family scan is plotted on a different row. Blue arrows indicate significant segregation distortion regions.

S15 Table). Rare allele load was negatively correlated with inbreeding depression for CUPR and EILN in maize ($r$ = -0.37, $r$ = -0.36, respectively), indicating that parents with more rare alleles overall had less difference between their outbred and inbred progeny, counter to the hypothesis that rare alleles tend to contribute to inbreeding depression for these traits.

We also measured relationships between inbreeding depression and proportions of rare alleles within RAS QTL intervals for the tested trait. In this case, we found significant positive correlations only for GW in maize ($r$ = 0.35, $p$ = 0.048) and total grain weight per plant (TGWP) in teosinte ($r$ = 0.30, $p$ = 0.037; Fig 8; S16 Table). Thus, we found a relationship between rare allele load within QTL intervals and parent-specific inbreeding depression only for one reproductive trait in teosinte only, and the correlation value was not very strong. Our inbreeding depression QTL are detected with limited genomic resolution, which hinders detection of relationships between predicted load within localized genomic regions and phenotypic inbreeding depression.

## Summary

The domestication bottleneck has resulted in reduced genetic diversity and a higher load of segregating predicted deleterious mutations in the global maize gene pool compared to teosinte [55,56,87]. Consistent with the reduced diversity of worldwide maize, we find that this particular pair of teosinte and maize populations shares 4.2 M segregating SNPs, with the maize population exhibiting 84.8% of the nucleotide diversity found in teosinte [44]. The site frequency spectra of the two populations reveal that maize is enriched for intermediate frequency derived alleles compared to teosinte (Fig 3). However, we detected fewer rare allele

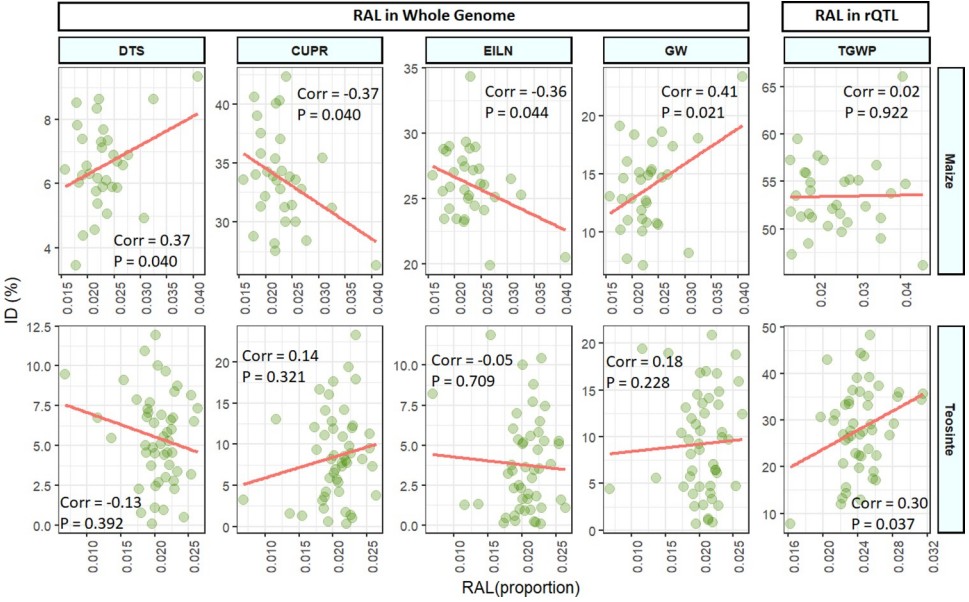

**Fig 8. Percent inbreeding depression of individual families related to the parental rare allele load (RAL) measured genome-wide and within QTL CI for affected traits in maize and teosinte.** Only traits for which a significant (P < 0.05) correlation was observed within one of the populations are displayed.

scan QTL and regions of significant segregation distortion in maize than in teosinte. The higher mean GERP score per site observed in teosinte presents an explanation: teosinte has more mutations of larger effect with lower allele frequencies, which would result in detecting more rare allele scan QTL in teosinte but less total inbreeding depression. In contrast, it appears that maize carries more common, smaller-effect mutations that cumulatively result in more inbreeding depression, of which a smaller proportion were detected as QTL.

The maize landrace studied here has a higher segregating genetic load predicted from evolutionary conservation of sequence variants, a higher frequency of derived alleles, and suffers more from inbreeding depression than its counterpart wild teosinte population collected from a nearby location. The genetic architecture of inbreeding depression in both maize and teosinte appears to be dominated by polygenic, small-effect recessive variants that individually are subjected to only limited selection pressure, but cumulatively contribute to substantial inbreeding depression (up to about 50% of seed yield in the maize population). In addition, rare larger-effect variants distributed among the parents of our study sample contribute a smaller but still substantial amount of genetic variation, as documented here as rare allele QTL effects, and these are more important in teosinte than in maize. Teosinte carries both more QTL previously detected by GWAS and also more rare QTL with large effects, many of which were not detected with a previous standard GWAS. Finally, an even rarer group of variants appears to be associated with juvenile lethality and are more important in teosinte than in maize. The more polygenic architecture of the segregating genetic load of our maize population renders it more recalcitrant to purging selection and poses a hindrance to modern breeding with landrace populations.

## Methods

### Population sampling, field evaluation, genotyping, and parentage inference

A sample of 70 teosinte plants was collected from a single field from the near the town of Palmar Chico in the Balsas river drainage of Mexico and 55 maize plants (classified as landrace Tuxpeño) plants were sampled from a single field nearby. Teosinte plants were selfed and intermated; 49 of these parental plants produced sufficient seed for progeny family evaluations (S1 and S2 Tables). Similarly, maize landrace plants were selfed and crossed, and 40 of the parental plants produced sufficient seed for progeny evaluations (S1 and S2 Tables).

Progenies were evaluated over two winter seasons under short daylengths (<12 h) in Homestead, Florida; details of the field experimental methods are given in Yang et al. [40]. Separate but nearby field blocks were used for teosinte and maize evaluations. Progenies were planted in randomized positions in the field blocks; at planting only the female parents of the progenies were known. Progenies and parents were genotyped using genotyping-by-sequencing (GBS) as described in Yang et al. [40]. The parentage of each progeny was determined using the GBS data of the parents and progeny plants, and this allowed us to distinguish selfed from outcrossed progeny. Eighteen traits were scored on both the teosinte and landrace progeny, these data were previously reported in Yang et al. [40] (Table 1).

GBS data were filtered for missing call rates, minor allele frequency, and imputed to obtain a total of 34,899 SNPs called for teosinte and 40,255 SNPs for maize landrace. The imputation method accounted for the parentage of each individual and the relatively high error rates for calling heterozygous sites that can occur with low coverage sequence data, as detailed in Yang et al. [40], and as implemented in ParentPhasingPlugin and ImputeProgenyStatesPlugin in TASSEL5 [https://bitbucket.org/tasseladmin/tassel-5-source/src/master/src/net/maizegenetics/analysis/imputation/; 88]. Briefly, the imputation involves phasing the parents using progeny information and inferring the parent state at each site using a hidden Markov

model. In addition to the SNP calls at each site, therefore, this imputation method calls the most likely parental haplotype inherited by each progeny at each site, using an arbitrary coding for parental haplotypes that is consistent across all selfed and outcrossed progenies of a particular parent. Following quality control of trait and SNP data, we retained data on 4,455 teosinte progenies and 4,398 maize progenies.

## Quantitative genetic analysis of inbreeding depression: Diallel model

We fitted a sequence of four diallel linear mixed models to the data for each trait and population [60,89]. The simplest model (model 1) was:

$$Y_{ijkl} = E_i + T_i + ET_{ij} + x_{sij}\beta_S + x_{Bij}B(E)_i + x_{Rij}\beta_{R1i} + x_{Rij}{}^2\beta_{R2i} + x_{Rij}{}^3\beta_{R3i} + x_{Rij}{}^4\beta_{R4i} + x_{Cij}\beta_{C1i} + x_{Cij}{}^2\beta_{C2i} + x_{Cij}{}^3\beta_{C3i} +$$
$$xR_{ij}{}^4\beta_{C4i} + M_{Ojk} + F_{Ojl} + M_{Sjk} + MF_{Ojkl} + ME_{Oijk} + FE_{Ojl} + ME_{Sijk} + MFE_{Oijkl} + \varepsilon_{ijkl}, \text{ where :}$$

$Y_{ijkl}$ is the observed phenotype on individual $j$ (with maternal parent $k$ and paternal parent $l$) in environment $i$.

Model 1 includes the following fixed effects:

$E_i$ is the effect of environment (year) $i$

$T_j$ is the effect of progeny type (selfed vs outcrossed) of individual $j$

$ET_{ij}$ is the interaction of progeny type and environment

$x_{ssij}$ is the deviation of the shading measurement on the $ij$th individual from the overall average shading measurement, described in Yang et al. [40]

$\beta_s$ is the effect of shading of neighboring plants

$x_{Bij}$ is a dummy variable indicating that individual $j$ in year $i$ is adjacent to a border row or a tractor tire track

$B(E)_i$ is the effect of border or tractor tire adjacent rows within year $i$

$x_{Rij}$ and $x_{Cij}{}^P$ are $p$ = first to fourth order polynomials of the deviation in the row and column directions, respectively, of the $ij$th plant's position from the center of the field in year $i$,

$\beta_{Rpi}$ and $\beta_{Cpi}$ are the regression coefficients associated with the $p$th polynomials for row and column trend effects within year $i$, respectively.

Model 1 also includes the following random effects:

$M_{Ojk}$ is the effect of the mother $k$ of progeny $j$ if $j$ is outcrossed; $M_{Ojk} \sim N(0, A_k\sigma_{GCA}^2)$, where $A_k$ is the matrix of pedigree relationships among the parents (some of the teosinte parents were themselves half-sibs), and $\sigma_{GCA}^2$ is the general combining ability (GCA) variance of the parents.

$F_{Ojl}$ is the effect of the father $i$ of progeny $j$ if $j$ is outcrossed. The effect of a particular father is constrained to be equal to the effect of the same parent as a mother ($F_{Ojl} = M_{Ojk}$ if $k = i$), so the distribution of father effects is equal to the distribution of mother GCA effects.

$M_{Sjk}$ is the effect of the mother $k$ of progeny $j$ if $j$ is selfed; $M_{Sj} \sim N(0, A_k\sigma_{S1}^2)$, where $\sigma_{S1}^2$ is the variance among selfed ($S_1$) families.

$MF_{Ojkl}$ is the interaction effect of the mother $k$ and father $l$ of progeny $j$ if $j$ is outcrossed; $MF_{Ojkl} \sim N(0, \sigma_{SCA}^2)$, where $\sigma_{SCA}^2$ is the specific combining ability (SCA) variation.

$ME_{Oijk}$, $FE_{Oijl}$, $ME_{Sijk}$, and $MFE_{Oijkl}$ are interactions between environment (year) and the previously defined mother and father effects. $ME_{Oijk}$ and $FE_{Oijk} \sim N(0, A_k \otimes I\sigma_{GCA*E}^2)$, where $A_k \otimes I$ is Kronecker product of the matrix of pedigree relationships among the parents ($A_k$) and $I$ is a $2 \times 2$ identity matrix, and $\sigma_{GCA*E}^2$ is the GCA-by-environment interaction variance. $ME_{Sijk} \sim N(0, A_k \otimes I\sigma_{S1*E}^2)$, where $\sigma_{S1*E}^2$ is the $S_1$ family-by-environment interaction variance. $MFE_{Oijkl} \sim N(0, \sigma_{SCA*E}^2)$, where $\sigma_{SCA*E}^2$ is SCA-by-environment variation.

$\varepsilon_{ijkl}$ is the residual effect; $\varepsilon_{ijkl} \sim N(0,\ \sigma^2_{\varepsilon i})$, where $\sigma^2_{\varepsilon i}$ is the residual variance for year $i$ (separate residual variances were fit for each year).

Model 2 includes all the same effects as model 1, but separates the residual variance for outcrossed and selfed progenies, so that four distinct residual variances were estimated: $\sigma^2_{\varepsilon O1}$ is the residual variance for outcross progeny in the first year, $\sigma^2_{\varepsilon O2}$ is the residual variance for outcross progeny in the second year, $\sigma^2_{\varepsilon S1}$ is the residual variance for selfed progeny in the first year, and $\sigma^2_{\varepsilon S2}$ is the residual variance for selfed progeny in the second year.

Model 3 includes all the same effects as model 2 and adds a covariance between the effects of a parent on outcrossed and selfed progenies:

$M_{Ojk}$ and $M_{Sjk}$ are jointly distributed as $N(0, A \otimes \Sigma)$, where $\sum = \begin{bmatrix} \sigma^2_{GCA} & \hat{\sigma}_{GCA,S1} \\ \hat{\sigma}_{GCA,S1} & \sigma^2_{S1} \end{bmatrix}$ and

$\hat{\sigma}_{GCA,S1}$ is the covariance between parent effects on outcrossed progeny (GCA) and selfed progeny.

Model 4 includes all the same effects as model 3 but adds reciprocal parental genetic effects:

$M_{ORjk}$ is the reciprocal effect of the mother $k$ of progeny $j$ if $j$ is outcrossed, it measures the difference between the effect of parent $k$ when used as female vs male. $M_{ORjk} \sim N(0,\ \sigma^2_{RGCA})$, where $\sigma^2_{RGCA}$ is the reciprocal general combining ability (GCA) variance of the parents.

$F_{ORjl}$ is the reciprocal effect of the father $l$ of progeny $j$ if $j$ is outcrossed.

$MF_{ORjkl}$ is the reciprocal effect that distinguishes the interaction of the mother $k$ and father $l$ progeny $j$ vs that interaction effect for the reciprocal cross if $j$ is outcrossed; $MF_{ORjkl} \sim N(0,\ \sigma^2_{RSCA})$, where $\sigma^2_{RSCA}$ is the reciprocal SCA variation.

$MFE_{ORjkl}$ is the interaction between environment (year) and the reciprocal SCA effect. $MFE_{ORjkl} \sim N(0,\ \sigma^2_{RSCA*E})$, $\sigma^2_{RSCA*E}$ is the reciprocal SCA-by-environment variation.

For every trait and population, all four models were fit using ASReml version 4.2 [90]. To obtain convergence with models 3 and 4, the covariance between GCA and S1 progeny effects was unconstrained, such that resulting estimates could correspond to correlations outside of the bounds of -1 to +1. Models were compared using Bayesian Information Criteria (BIC). From selected models, the following parameters were estimated [62,91]:

$$\hat{\sigma}^2_A = 4\hat{\sigma}^2_{GCA}$$

$$\hat{\sigma}^2_D = 4\hat{\sigma}^2_{SCA}$$

$h^2 = \frac{\hat{\sigma}^2_A}{0.5\hat{\sigma}^2_A + 0.25\hat{\sigma}^2_D + 2\hat{\sigma}^2_{GCA*E} + \hat{\sigma}^2_{SCA*E} + \hat{\sigma}^2_{\varepsilon 0}}$, where $\hat{\bar{\sigma}}^2_{\varepsilon 0}$ is the average residual variance for outcross progenies from each year. The residual variance includes both within-full-sib-family genetic variance $(0.5\hat{\sigma}^2_A + 0.75\hat{\sigma}^2_D)$ and experimental error variance.

The marginal mean values of each progeny type (outcrossed and selfed) were predicted at common values of the fixed effects. The coefficient of inbreeding depression was estimated as the proportional change in population mean between outcrossed and selfed progenies: $\delta = 1 - \frac{outcrossed\ mean}{selfed\ mean}$ [92].

The genetic variance among selfed ($S_{0:1}$) families is expected to be $\sigma^2_A + 0.25\sigma^2_D + \sigma_{ADI} + 0.125\sigma^2_{DI}$, ignoring epistatic variances, where $\sigma_{ADI}$ is the covariance between additive and dominance effects in inbred individuals $(2\sum_{k=1}^{n} E[\alpha_{ki}\delta_{kii}])$ and $\sigma^2_{DI}$ is the variance of dominance effects in inbred individuals $(\sum_{k=1}^{n} E[\delta^2_{kii}] - (E[\delta_{kii}])^2$ [61,93]. Further, the expectation of the covariance between parental GCA values and their selfed progenies is $Cov(GCA,\ S1) = 0.5\sigma^2_A + 0.25\sigma_{ADI}$ [30]. Therefore, we can estimate $\sigma_{ADI}$ as $4(Cov(GCA,\ S1) - 0.5\hat{\sigma}^2_A)$. Similarly, we can estimate $\sigma^2_{DI}$ as

$8(\hat{\sigma}_{S1}^2 - 4(Cov(GCA, S1)) + \hat{\sigma}_A^2 - 0.25\hat{\sigma}_D^2)$. Parameter estimates were considered to be significantly different from zero if their absolute value exceeded twice their standard error.

### Regression of trait values on genomic inbreeding coefficient estimate

The genomic realized estimate of the inbreeding coefficient ($F$) of each individual was obtained from the realized additive genomic relationship matrix as 1 minus the diagonal elements [94]. This estimate of the inbreeding coefficient was included in a separate linear mixed model that contained the same fixed covariates (except for $T_j + ET_{ij}$) and the same residual effects distribution as diallel model 1 and adding the genomic estimated inbreeding coefficient and its interaction with environments as fixed covariates. In addition, the model included these random effects:

$Aij$, the polygenic additive effect of the $ij$th plant, with distribution $A_{ij} \sim N(0, A\sigma_A^2)$, where **A** is the realized additive genomic relationship matrix and $\sigma_A^2$ is the additive genetic variance, $D_{ij}$, is the polygenic dominance effect of the $ij$th plant, with distribution $D_{ij} \sim N(0, D\sigma_D^2)$, where **D** is the realized additive genomic relationship matrix and $\sigma_D^2$ is the dominance genetic variance, $G{\times}E_{ij}$ is the interaction of polygenic effect of the $j$th plant in environment $i$, with distribution $G{\times}E_{ij} \sim N(0, A_1 \oplus A_2\sigma_{AE}^2)$, where $A E_i$ is the realized additive relationship matrix for individuals grown in year $i$ and $(A_1 \oplus A_2)$ is a block-diagonal structure that includes non-zero covariances between plants grown in the same year, but zero covariance between plants grown in different years.

This model was fit with ASReml version 4.2, and the regression effect of the inbreeding coefficient was tested for significance using a conditional F-test. The slope of regression on the inbreeding coefficient was estimated as its main effect plus the average of its interactions with years.

The influence of epistasis on inbreeding depression can be tested by including the squared inbreeding coefficient as an additional regression variable [4,62]. We computed the square of the genomic estimated inbreeding coefficient (fixing the squared value of the slightly negative estimates for some individuals as zero) and fit this coefficient and its interaction with years in a second model otherwise identical to the previous model.

### Rare allele scan (RAS) to detect private haplotype effects

To detect large-effect rare variants that are carried by only one parent, we developed a novel linkage scan that tests each genetic map position for the effect of one parental allele compared to all other parental alleles (Fig 1). This analysis relies on progeny genotype calls that represent identity-by-descent parental allele calls that were generated during the imputation and phasing of the GBS data described in detail in Yang et al. [40]. These genotype calls reflect the phased parent haplotypes inherited by each individual progeny at each SNP.

The first step of the rare allele scan (RAS) was to create separate linkage maps for each population. Recombinations of parental haplotypes observed in the progeny were used to estimate linkage distances between SNPs. Within each interval of 1 cM we expect to observe that 1% of the progeny haplotypes have recombined compared to their parents. We selected SNP markers at 1 cM intervals to represent the linkage maps by identifying marker pairs separated by 0.01* (2N) recombinations, where N is the genotyped progeny sample size. The resulting linkage maps had total distances of 1371 and 1426 cM for teosinte and maize, respectively.

In a second step, were created a new data set for each parental haplotype, in which the progeny haplotype calls were recoded to the number of copies of one parental haplotype carried by each progeny at each of the linkage map markers (Fig 1). Then, we fit a linear model for each

combination of parental haplotype and linkage marker:

$$Y_{ij} = x_{haij}a + x_{hdij}d +$$
$$E_i + F_{ij}\beta_{Fi} + x_{sij}\beta_S + x_{Bij}B(E)_i + x_{Rij}\beta_{R1i} + x_{Rij}^2\beta_{R2i} + x_{Rij}^3\beta_{R3i} + x_{Rij}^4\beta_{R4i} + x_{Cij}\beta_{C1i} + x_{Cij}^2\beta_{C2i} + x_{Cij}^3\beta_{C3i} + xR_{ij}^4\beta_{C4i} +$$
$$PC1_{ij} + PC2_{ij} + \ldots + PC50_{ij} + \varepsilon_{ij}, \text{ where :}$$

$x_{haij}$ is the number of the parental haplotype alleles carried individual $j$ within year $i$ at the tested map position

$a$ is the additive effect of the parental allele at the tested map position

$x_{hdij}$ is an indicator for heterozygosity for the current parental haplotype allele in individual $j$ within year $i$ at the tested map position

$d$ is the dominance effect of the parental allele at the tested map position

$E_i$ is the effect of environment (year) $i$

$F_{ij}$ is the genomic estimated inbreeding coefficient of individual $j$ within year $i$

$\beta_{Fi}$ is the effect of inbreeding within year $i$

$PC1_{ij} + PC2_{ij} + \ldots + PC50_{ij}$ are the scores for individual $j$ within year $i$ on each of the first 50 principal components of the genome-wide SNP data, previously computed by Chen et al. [44]

$E_i$, $x_{sij}$, $\beta_s$, $x_{Bij}$, $B(E)_i$, $x_{Rij}^p$, $x_{Cij}^p$, $\beta_{Rpi}$, and $\beta_{Cpi}$ are as defined in the diallel model previously.

Models were fit using ordinary least squares, implemented in the statsmodels package of Python [95]. The LOD score for each marker-parent haplotype combination was calculated as:

LOD = $(n/2)\log_{10}(RSS0/RSSmark)$, where $n$ is the progeny sample size, RSSmark is the residual sum of squares of the full model shown above, and RSS0 is the residual sum of squares from a model all of the effects in the full model except for the marker additive and dominance effects.

This model is equivalent to the genome-wide association test implemented by Chen et al [44] except that it tests the additive and dominance effects of a single parental allele defined by identity-by-descent instead of the usual additive and dominance effects of SNP alleles defined by identity-in-state (Fig 1).

Initial putative RAS quantitative trait loci (QTL) positions were declared at marker-haplotype combinations with LOD scores greater than or equal to 4.0. LOD-2 support intervals for the initial QTL scans for each parental haplotype were defined based on the results of the initial linkage scan as contiguous regions of the genetic map LOD scores within 2 LOD of the largest effect in the region.

The final step of this analysis was a forward stepwise regression fitting the model above for the most significant marker-haplotype and then adding each initially declared QTL to the model, one at a time. The QTL that improved (decreased) the Bayesian Information Criterion (BIC) of the model most was added to the model. This process was repeated until no more QTL improved the model BIC. Only QTL remaining in the final model were reported as significant QTL. QTL number, effects, individual variances, and combined variances were estimated from this final model.

## Testing for non-random clustering of RAS QTL and GWAS association positions

Physical positions of linkage map markers were located on the maize B73 AGP version 4 reference sequence [96], as were SNP markers used for GWAS previously by Chen et al. [44]. This allowed estimation of the physical sequence size and local recombination rate (cM per Mbp) for RAS QTL and identification of previously-identified trait-associated SNPs inside of QTL

for the same trait. For each trait, the proportion of RAS QTL containing associated SNPs identified by GWAS, the proportion of QTL overlapping within a population (because they were detected in different parents), and the proportion overlapping between populations were computed using the 2-LOD support intervals to define QTL windows. The distributions of the proportion of QTL containing GWAS associations or QTL overlapping within or between populations by chance were simulated by a permutation test. A single permutation of QTL positions was made by randomly assigning windows corresponding to the cM size of the observed QTL support intervals for a trait to the linkage map. The proportion of GWAS associations within these permuted windows represents one realization of the proportion of overlap between GWAS and QTL positions under the null hypothesis of random QTL positions with respect to GWAS associations. Similarly, the proportion of QTL positions overlapping within a population within one permuted data set represented the proportion of overlapping QTL among parents under the null hypothesis of random distribution of QTL positions among parents. Finally, to estimate the proportion of overlapping QTL between maize and teosinte for a common trait, QTL window positions were permuted within each population and compared. We repeated this process 5000 times for each set of comparisons. This provided an empirical estimate of the distribution of the proportion of overlaps under the null hypothesis of random QTL positions. Two-sided empirical $p$-values for the observed proportion of QTL overlaps were estimated based on the percentile of the permuted distribution corresponding to the observed proportion of overlaps for a given comparison.

### Estimation of inbreeding depression associated with rare allele scan QTL and genome-wide associated SNPs

We predicted the amount of population inbreeding depression expected to be caused by a single RAS QTL as $2Fpqd$[62], where $F$ is the inbreeding coefficient, and $p$ and $q$ are the major and minor allele frequencies. W set $F = 0.5$ for the one generation of selfing used in this experiments, and $q = 1/2N$ for N parents because we model the RAS QTL alleles as carried on only one haplotype of one parent, and $d$ is the estimated dominance effect of the QTL scaled to the outcross progeny mean rather than to the phenotypic standard deviation so that we can compare this to the observed percent inbreeding depression observed in selfed versus outcrossed progeny means computed from the diallel analysis. The predicted amount of inbreeding depression associated with each RAS QTL was summed over QTL for a given trait and population and compared to the observed inbreeding depression. A similar comparison was made for GWAS associations, using the minor allele frequency at each QTL observed in the parents as $q$.

### Linkage scan for segregation distortion

For both maize LR and teosinte, we used the same consensus linkage maps used in the private haplotype test described above to identify genomic regions exhibiting segregation distortion. Selfed families with 20 or more progenies were included in this analysis. We analyzed 24 and 42 selfed families for maize landraces and for teosinte, respectively. The number of progenies per selfed family analyzed ranged from 20 to 125 (mean 52.9) in maize and from 20 to 95 (mean 45.2) in teosinte. In total, 1,271 selfed progenies were included in this analysis in maize and 1,899 in teosinte. Chi-square tests of goodness of fit of the observed segregation ratios at each linkage map marker to the expected 1:2:1 proportions were performed with the R package OneMap [97].

Multiple test correction for the purpose of declaring genome-wide significance thresholds is challenging in genetic studies because of the complex correlation structure among marker tests. We inferred the effective number of independent tests (ENT) based on the pairwise LD

between SNPs [98]. Matrices of correlation coefficients between each pair of SNPs in the consensus linkage map of each chromosome were used to compute ENT for each chromosome using the R package poolR [99]. The chromosome-specific ENTs were summed to obtain a genome-wide ENT, and the significance threshold for segregation distortion was set as 0.05/$ENT_{total}$. Significance thresholds were computed separately for maize and teosinte. Furthermore, we required a contiguous group of at least three consecutive markers passing the whole genome threshold to declare a significant segregation distortion region (SDR).

Finally, to ensure that putative significant SDR were not an artefact of the imputation and phasing procedure, we performed a check on the segregation ratios of genotype calls in the raw GBS data. The first step was to delimit windows corresponding to each SDR. In a second step, we selected the raw GBS markers that passed the quality control filters imposed before imputation described by Yang et al. (2019) within the SDR windows for the specific parent of the family where segregation distortion was observed and that were heterozygous in the parent. Then we obtained the P-values for the chi-square tests for goodness of fit to the expected 1:2:1 segregation ratio in the progenies of the affected family. We required more than half of the informative raw GBS markers to have significant P-values (P < 0.05) for these chi-square tests to confirm the SDR originally identified with the imputed data. We calculated the recombination rate in each significant SDR as cM length of the region divided by the physical length of the region in Mbp. For comparison, we computed both the genome-wide average recombination rate per Mbp and also the average recombination rate specific to the chromosome on which the SDR was located.

## The relationship between parent-specific rare allele load and inbreeding depression

Genotypes at 20,631,107 and 17,819,152 SNPs were obtained for each parent of the maize and teosinte, populations, respectively [44]. To compute the genome-wide rare allele load of each parent, we first subset the complete set of SNPs to include only those with minor allele frequencies < 0.05 (considered rare in our study). After this filter 9,444,274 and 9,823,305 markers were retained for maize and teosinte, respectively. Each parent's rare allele load was the mean frequency of rare alleles across these loci. We computed the correlation between each parent's rare allele load and parent-specific percent inbreeding depression for each trait within each population. We included both homozygous and heterozygous rare allele calls in this computation (with homozygous calls given double weight) because inbreeding depression was measured as the difference between each parent's outbred and inbred progeny values, not as the difference between the parent itself and its selfed progeny. Thus, although homozygous recessive deleterious alleles in the parent do not contribute to a difference between the parent and its inbred progeny, they do contribute to a difference between inbred progeny and the outbred progenies of that parent (as they will be almost entirely complemented by common alleles from other parents in outbred progenies). Next, we computed the rare allele load carried by each parent at the subset of genomic positions within the support intervals of all rare allele scan QTL for a given trait. For each trait, correlations were estimated between the proportion of inbreeding depression exhibited by each parent and the parent-specific QTL rare allele load.

## Estimation of mutational burden per parent

Genetic load for each parent was estimated from genomic evolutionary rate profiling (GERP) scores using GERP scores computed by Kistler et al. (2018) from whole genome sequence data. Only sites segregating in the respective population (maize or teosinte) with an allele call in sorghum as outgroup, and a positive GERP score were included. Using the sorghum allele as

ancestral allele, per individual burden was calculated under a partially recessive model with a per site burden equal to the GERP score for that site for homozygous derived genotypes, $0.25 \times$ GERP for heterozygous genotypes and 0 for homozygous ancestral genotypes. From this, the total and mean GERP effects across loci were computed per individual.

## Supporting information

**S1 Fig. Percent inbreeding depression, regression coefficient of individual trait values on individual genomic inbreeding estimates, and ratio of dominance to total genetic variance for 18 traits measured in maize and teosinte.**
(PNG)

**S2 Fig. Mean value of self-fertilized progenies as a proportion of outbred progeny means for 18 traits in teosinte and maize.**
(PNG)

**S3 Fig. Predicted segregating mutational burden per parent in the parents of maize and teosinte populations.** (A) Total segregating deleterious sites per individual parent, (B) homozygous deleterious sites per parent, (C) Heterozygous deleterious sites per parent, (D) Mean burden per site per parent based on genomic evolutionary rate profiling (GERP) score under a model of partial recessivity, including only sites segregating within the parent's population, (E) Mean homozygous burden per parent, (F) Mean heterozygous burden per parent, (G) Total burden per parent based on GERP scores, (H) Total homozygous burden per parent based on GERP scores, (I) Total heterozygous burden per parent.
(PNG)

**S4 Fig. Distribution of genomic inbreeding coefficient estimates among individuals in maize and teosinte populations.**
(PNG)

**S5 Fig. Rare allele scan results for vegetative and flowering traits in maize.** Each row of figures corresponds to one trait. Each column corresponds to one of the ten chromosome pairs in maize. Logarithm of odds (LOD) scores for QTL models are plotted for the 2-LOD support interval for each QTL. LOD curves correspond to the effects of a single parental haplotype, and different parent effects are plotted with different colors. Blue dots represent the -log10 *p*-values of significant GWAS associations from Chen et al. [44].
(PNG)

**S6 Fig. Rare allele scan results for environmental response traits in maize.** Each row of figures corresponds to one trait. Each column corresponds to one of the ten chromosome pairs in maize. Logarithm of odds (LOD) scores for QTL models are plotted for the 2-LOD support interval for each QTL. LOD curves correspond to the effects of a single parental haplotype, and different parent effects are plotted with different colors. Blue dots represent the -log10 *p*-values of significant GWAS associations from Chen et al. [44].
(PNG)

**S7 Fig. Rare allele scan results for reproductive traits in maize.** Each row of figures corresponds to one trait. Each column corresponds to one of the ten chromosome pairs in maize. Logarithm of odds (LOD) scores for QTL models are plotted for the 2-LOD support interval for each QTL. LOD curves correspond to the effects of a single parental haplotype, and different parent effects are plotted with different colors. Blue dots represent the -log10 *p*-values of

significant GWAS associations from Chen et al. [44].
(PNG)

**S8 Fig. Rare allele scan results for vegetative and flowering traits in teosinte.** Each row of figures corresponds to one trait. Each column corresponds to one of the ten chromosome pairs in maize/teosinte. Logarithm of odds (LOD) scores for QTL models are plotted for the 2-LOD support interval for each QTL. LOD curves correspond to the effects of a single parental haplotype, and different parent effects are plotted with different colors. Blue dots represent the -log10 $p$-values of significant GWAS associations from Chen et al. [44].
(PNG)

**S9 Fig. Rare allele scan results for environmental response traits in teosinte.** Each row of figures corresponds to one trait. Each column corresponds to one of the ten chromosome pairs in maize/teosinte. Logarithm of odds (LOD) scores for QTL models are plotted for the 2-LOD support interval for each QTL. LOD curves correspond to the effects of a single parental haplotype, and different parent effects are plotted with different colors. Blue dots represent the -log10 $p$-values of significant GWAS associations from Chen et al. [44].
(PNG)

**S10 Fig. Rare allele scan results for reproductive traits in teosinte.** Each row of figures corresponds to one trait. Each column corresponds to one of the ten chromosome pairs in maize/teosinte. Logarithm of odds (LOD) scores for QTL models are plotted for the 2-LOD support interval for each QTL. LOD curves correspond to the effects of a single parental haplotype, and different parent effects are plotted with different colors. Blue dots represent the -log10 $p$-values of significant GWAS associations from Chen et al. [44].
(PNG)

**S11 Fig. Proportion ($R^2$) of variation associated with genetic background (parentage estimated by principal components of the genome-wide marker data) and rare allele scan QTL effects for vegetative and flowering traits in maize and teosinte.** Full model includes parentage, QTL additive, and QTL dominance effects. The proportion of variance due specifically to parentage, additive plus dominance QTL effects, additive QTL effects only, or dominance QTL effects only was estimated by measuring the decrease in $R^2$ after removing one of those factors from the full model.
(PNG)

**S12 Fig. Proportion ($R^2$) of variation associated with genetic background (parentage estimated by principal components of the genome-wide marker data) and rare allele scan QTL effects for environmental response traits in maize and teosinte.** Full model includes parentage, QTL additive, and QTL dominance effects. The proportion of variance due specifically to parentage, additive plus dominance QTL effects, additive QTL effects only, or dominance QTL effects only was estimated by measuring the decrease in $R^2$ after removing one of those factors from the full model.
(PNG)

**S13 Fig. Proportion ($R^2$) of variation associated with genetic background (parentage estimated by principal components of the genome-wide marker data) and rare allele scan QTL effects for reproductive traits in maize and teosinte.** Full model includes parentage, QTL additive, and QTL dominance effects. The proportion of variance due specifically to parentage, additive plus dominance QTL effects, additive QTL effects only, or dominance QTL effects only was estimated by measuring the decrease in $R^2$ after removing one of those factors from

the full model.
(PNG)

**S14 Fig. Number of rare allele scan QTL detected per trait group and population (top row).** Number of large-effect ($>$ 1 phenotypic standard deviation effect) rare allele scan QTL detected per trait group and population.
(PNG)

**S15 Fig. Relationships between the number of rare allele scan QTL detected per parent and the progeny sample size of each parent for teosinte and maize.**
(PNG)

**S16 Fig. Proportion of traits with significant clustering of QTL between maize and teosinte populations, between QTL and significant GWAS associations, and between QTL detected in different parents within populations.**
(PNG)

**S17 Fig. Relationship between additive (red) and dominance (blue) effect estimates associated with rare allele scan (RAS) QTL (X-axis) and corresponding GWAS associations inside the QTL intervals (Y-axis) in maize and teosinte.** Effects are standardized to the phenotypic standard deviation for each trait and population.
(PNG)

**S18 Fig. Relationship between standardized additive and dominance effect estimates of rare allele scan QTL.** Effects are standardized to the phenotypic standard deviation for each trait and population.
(PNG)

**S19 Fig.** Standardized additive ($a$) and dominance ($d$) effects of rare allele scan QTL effects plotted by genome position within each of the ten chromosome pairs for both maize (A) and teosinte (B). Effects are standardized to the phenotypic standard deviation for each trait and population. Effects are plotted within trait categories. Right hand panels ("Eff_Direct") show the proportion of QTL allele effects where the rare variant effect is favorable (against the direction of inbreeding depression) or unfavorable (in the same direction as inbreeding depression) within categories based on the level of dominance of the rare allele, summed over all traits. Favorable allele effects are (partially to fully) recessive when $a > 0$ and $d < 0$, (partially to fully) dominant when $a > 0$ and $d < 0$, and overdominant when $d > a > 0$. Unfavorable alleles are (partially to fully) recessive when $a < 0$ and $d > 0$, (partially to fully) dominant when $a < 0$ and $d < 0$ alleles, and overdominant when $d < a < 0$.
(PNG)

**S20 Fig. Proportion of trait inbreeding depression predicted based on rare allele scan (RAS) or genome-wide association study (GWAS) QTL dominance effect estimates and allele frequencies ($\sum_i^N p_i(1 - p_i)d_i$) for each trait in maize and teosinte.**
(PNG)

**S21 Fig. Segregation distortion regions (SDR; blue bars) superimposed on physical linkage maps of teosinte chromosomes also carrying seed dormancy QTL (green diamonds) detected by Chen et al. [44].** The local recombination rate (cM/Mb) is plotted as intensity of red color for each 1-cM window within either population. The position of *gametophyte factor 2* (*Ga2*) locus is indicated in purple (this locus has not been finely mapped to date).
(PNG)

**S22 Fig. Observed selfed progeny genotype frequencies at loci detected as segregation distortion regions in maize and teosinte.** Genotypes *aa* and *bb* refer to homozygotes for one of the parental alleles, *ab* refers to heterozygotes. Shapes refer to particular families.
(PNG)

**S23 Fig. Observed selfed progeny genotype frequencies at loci detected as segregation distortion regions and overlapping known gametophyte factors in teosinte.** Genotypes *aa* and *bb* refer to homozygotes for one of the parental alleles, *ab* refers to heterozygotes. Shapes refer to particular families. Right hand bar indicates the position of SDR (blue bars) and gametophyte factors (purple) on chromosomes 4 and 5, with the specific parents giving rise to the SDR indicated within their SDR.
(PNG)

**S24 Fig.** Segregation distortion regions (SDR) superimposed on physical linkage maps of maize (A) or teosinte (B). The local recombination rate (cM/Mb) is plotted as intensity of red color for each 1-cM window within either population. The positions of *gametophyte factor 1* (*Ga1*), *gametophyte factor 2* (*Ga2*), and *teosinte crossing barrier 1* (*Tcb1*) loci are indicated in purple.
(PNG)

**S1 Table. Teosinte mating design details.** The number of progenies from each maternal parent are in rows, the number of progenies from each paternal parent are in columns. Values on the diagonal correspond to the number of selfed progenies per parent.
(XLSX)

**S2 Table. Maize mating design details.** The number of progenies from each maternal parent are in rows, the number of progenies from each paternal parent are in columns. Values on the diagonal correspond to the number of selfed progenies per parent.
(XLSX)

**S3 Table. Inbreeding depression summary results from diallel analysis presented in this paper and realized relationship matrix analyses presented by Yang et al (2019) for both teosinte and maize.** Overall outbred progeny mean values (S0_mean), selfed progeny mean values (S1_mean), scaled S1 progeny means as a proportion of the S0 mean value (S1_mean_-scaled), percent inbreeding depression ((S1_mean–S0_mean)/S0_mean; ID_perc), absolute value of percent inbreeding depression (ID_perc_abs), the slope of the regression of individual trait values on estimated genomic inbreeding coefficient (Beta), the absolute value of the regression slope as a proportion of outbred mean value (Beta_scaled), additive genetic variance estimated from diallel (Additive), additive genetic variance estimated from the realized relationship matrices (VA_realized), dominance genetic variance estimated from the diallel (Dominance), dominance genetic variance estimated from the realized relationship matrices (VD_realized), narrow-sense heritability estimated from the diallel (h2), narrow-sense heritability estimated from the realized relationship matrices (h2_realized), ratio of dominance to total genetic variation estimated from the diallel (D_G_ratio), ratio of dominance genetic variance to total phenotypic variance estimated from the diallel (D_P_ratio), ratio of dominance to total genetic variation estimated from the realized relationship matrices (D_G_ratio_realized), ratio of dominance genetic variance to total phenotypic variance estimated from the realized relationship matrices (D_P_ratio_realized), heritability of S1 family means (H_S1), error variance estimated from outbred progenies (ErrorS0), error variance estimated from selfed progenies (ErrorS1), ratio of S1 to S0 error variances (S1_S0_Resid), total phenotypic variance among outbred individuals (Pheno_out), total phenotypic variance among selfed

individuals (Pheno_S1), genotypic variance among S1 families (S1_Var), ratio of observed S1 family genetic variance to expected S1 family variance estimated as $\sigma_A^2 + 0.25\sigma_D^2$, based on additive and dominance variances estimated from outbred progenies, correlation between out-bred and selfed breeding values for each parent (rS0S1), correlation between outbred breeding value and inbreeding depression for each parent (rS0BV_ID), correlation between outbred breeding value and percent inbreeding depression for each parent (r_S0BV_IDperc).
(CSV)

**S4 Table. Genetic burden per parent by predicted deleterious genotypic class: parental code (individual), population, genotypic count at deleterious sites (anc_geno; 0 = homozygous for ancestral allele, 1 = heterozygous, 2 = homozygous for derived, putatively deleterious allele), number of sites within genotypic class (del_sites), mean burden score per site within genotypic class (mean_load), sum of load scores across sites within genotypic class (total_load).**
(CSV)

**S5 Table. Outbred and inbred progeny mean values, and family-specific inbreeding depression for individual parents within maize and teosinte populations.** S0_effect and S1_effect are family effects scaled to the overall outbred population mean.
(CSV)

**S6 Table. Estimates, their standard errors, and significance of quantitative genetic parameters estimated from the diallel model by population and trait: dominance and additive variance, the proportion of observed S1 family variance to the expected value based on additive and dominance variance components alone, the covariance of additive and dominance effects of inbred individuals ($\sigma_{ADI}$, or 'D1'), the variance of dominance effects of inbred individuals ($\sigma_{DI}^2$ or 'D2'). Significance refers to absolute values of estimates greater than twice their standard error for all parameters except the proportion of observed to expected S1 family variance, where significance is declared if the estimate plus or minus twice its standard error does not encompass 1.**
(CSV)

**S7 Table. Rare allele scan QTL effects in teosinte and maize.** SNP, parent name, and haplotype number for QTL peak joined together (Marker), proportion of trait variance due to additive effect of this haplotype vs. all others (r2_A), proportion of trait variance due to dominance effect of this haplotype (r2_D), proportion of trait variance due to the combined additive and dominance effects of this haplotype vs. all others (r2_AD), additive effect of this haplotype allele vs. all others (a_effect), p-value associated with the additive effect (a_pval), t-value associated with the additive effect (a_tval), dominance effect of this haplotype (d_effect), p-value associated with the dominance effect (d_pval), t-value associated with the dominance effect (d_tval), number of heterozygotes for this haplotype (N_het), number of homozygotes for this haplotype (N_hom), peak SNP, parent, haplotype number, chromosome, physical position of SNP on AGP version 4 of the B73 reference sequence (pos_Agpv4), cM position of SNP, SNP defining left side of QTL interval (Pos_start), genetic position of SNP defining left side of QTL support interval (cM_start), physical position of SNP defining left side of QTL support interval (Pos_start), SNP defining right side of QTL interval (SNP_end), genetic position of SNP defining right side of QTL support interval (cM_end), physical position of SNP defining right side of QTL support interval (Pos_end), population, trait group, and number of GWAS associations identified by Chen et al. [44] for same trait and population inside the support interval (N. GWAS.hits).
(CSV)

**S8 Table. Pleiotropic rare allele scan QTL in maize.** Filtered version of S7 Table including only overlapping haplotype QTL positions with effects on more than one trait in maize. (XLSX)

**S9 Table. Pleiotropic rare allele scan QTL in teosinte.** Filtered version of S7 Table including only overlapping haplotype QTL positions with effects on more than one trait in teosinte. (XLSX)

**S10 Table. Overlapping rare allele scan QTL and GWAS associations.** GWAS associations detected by Chen et al. [44] encompassed by rare allele scan QTL for the same trait and population. SNP, parent name, and haplotype number for rare allele scan QTL peak joined together (QTL), marker identified by GWAS (SNP), chromosome (chrom_QTL and chrom_GWAS), additive effect of GWAS association (a_effect_SNP), dominance effect of GWAS association (d_effect_SNP), AgpV4 physical position of GWAS associated SNP (Position), Std. Dev of trait within the population, and proportion of trait variance associated with GWAS SNP (r2_GWAS). Other trait columns are the same as for S8 Table. (CSV)

**S11 Table. Summary of overlapping rare allele QTL and GWAS associations.** Comparisons include QTL between populations, which are overlapping QTL for the same trait mapped in both maize and teosinte; QTL within populations, which are overlapping QTL for the same trait and population, but identified as the contrast between different specific parental alleles and the rest of the population alleles; GWAS hits within QTL, which are GWAS SNPs that are encompassed by a QTL for the same trait and population; and QTL containing GWAS SNPs, which are QTL that encompass a GWAS SNP for the same trait and population. Each comparison is quantified by the mean number of overlaps per comparison (Mean.overlaps), the number of comparisons with zero overlaps (N.zero), the empirical permutation-based *p*-value for mean overlaps (Means.overlaps.pval), the empirical permutation-based p-value for the number of comparisons with zero overlaps (N.zero.overlaps.pval), and the total number of QTL or GWAS associations that represent the 'denominator' of these comparisons (N.QTL.or.hits). In addition, a Boolean variable (Clustered) indicates whether the observed mean number of overlaps is greater than the mean number obtained from the permutation test; this is not a significance test, if the empirical *p*-value indicates that the observed number of overlaps is significantly different than the random distribution, the variable Clustered indicates evidence of clustering (True) or overdispersion (False). (CSV)

**S12 Table. Proportions of overlapping rarel allele scan and GWAS associations summarized by population and trait group: Nooverlap and Overlap are the numbers of QTL that encompass GWAS associations summed over traits within a trait group; Overlap.perc is the proportion of QTL that encompass a GWAS association; N.sign.mean.overlap and Perc.sign.mean.overlap are the total number and percentage of traits within the trait group for which the overlapping proportion is significantly greater than expected based on random distributions; N.sign.N.zero and Perc.sign.N.zero are the total number and percentage of traits within the trait group for which number of QTL with zero GWAS associations is greater than expected based on random distributions.** (CSV)

**S13 Table. Mean correlation between predicted and observed phenotypic values, and mean proportion of variance associated with models trained in test sets and evaluated in disjoint**

**validation sets over ten folds of cross-validation of rare allele scan QTL models.** In each fold, the forward selection algorithm for the rare allele scan QTL was performed in a training set of 90% of the population individuals and the resulting model was used to predict the trait values of the held-out 10% validation set of individuals. The correlation between predicted and observed values was estimated from the null model containing only environmental covariates but no genetic effects (r_null), a model adding principal components of the genome-wide marker data (r_pc), a model adding selected rare allele scan QTL effects (r_pc), and a model including both principal components and QTL (r_full). The change in correlation from adding QTL to the null model (r_qtl_vs_null) and the change in correlation from adding QTL to the PC model (r_pc_vs_qtl) was measured for each trait. Similar values are estimated for the proportion of trait variation explained ($R^2$) in the validation set for each model.
(CSV)

**S14 Table. Summary of segregation distortion analysis in maize and teosinte, indicating the parent of the self-fertilized family where segregation distortion was detected (Family); the chromosome (Chr); the SNP name (marker_start), cM position (cM_start), AgpV4 position (pos_start(bp) and pos_start(Mb)) of the SNP defining the left side of the segregation distortion region; the SNP name (marker_end), cM position (cM_end), AgpV4 position (pos_end(bp) and pos_end(Mb)) of the SNP defining the right side of the segregation distortion region; the length of the region in bases (Length(Mb)) and in cM (Length(cM)), the recombination rate within the region (cM/Mbp; SDR-RR), and the average recombination rate in the chromosome (CRR).**
(XLSX)

**S15 Table. Correlations between rare allele load across the whole genome and inbreeding depression for 18 traits in maize and teosinte.**
(XLSX)

**S16 Table. Correlations and p-values between rare allele load within QTL regions and inbreeding depression for 18 traits in maize and teosinte.**
(XLSX)

## Acknowledgments

We thank Jason Brewer for his assistance with the field evaluations for this project.

## Author Contributions

**Conceptualization:** Edward S. Buckler, John F. Doebley, James B. Holland.

**Data curation:** Luis Fernando Samayoa, Bode A. Olukolu, Chin Jian Yang, Qiuyue Chen.

**Formal analysis:** Luis Fernando Samayoa, Chin Jian Yang, Qiuyue Chen, Markus G. Stetter, Peter J. Bradbury, Qi Sun, Jinliang Yang, James B. Holland.

**Investigation:** Luis Fernando Samayoa, Bode A. Olukolu, Chin Jian Yang, Qiuyue Chen, Alessandra M. York, Maria Cinta Romay.

**Project administration:** Jeffrey C. Glaubitz, Maria Cinta Romay.

**Resources:** Jose de Jesus Sanchez-Gonzalez, Jeffrey C. Glaubitz, Qi Sun.

**Software:** Peter J. Bradbury.

**Supervision:** Jeffrey Ross-Ibarra, Edward S. Buckler, John F. Doebley, James B. Holland.

**Writing – original draft:** Luis Fernando Samayoa, James B. Holland.

**Writing – review & editing:** Bode A. Olukolu, Chin Jian Yang, Qiuyue Chen, Markus G. Stetter, Alessandra M. York, Jose de Jesus Sanchez-Gonzalez, Peter J. Bradbury, Maria Cinta Romay, Qi Sun, Jinliang Yang, Jeffrey Ross-Ibarra, Edward S. Buckler, John F. Doebley.

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
