## [Decision Letter · Decision Letter 0]

1 Oct 2021

Dear Dr Holland,

Thank you very much for submitting your Research Article entitled 'Domestication Reshaped the Genetic Basis of Inbreeding Depression in a Maize Landrace Compared to its Wild Relative, Teosinte' to PLOS Genetics.

The manuscript was fully evaluated at the editorial level and by independent peer reviewers. The reviewers appreciated the attention to an important problem, but raised some substantial concerns about the current manuscript. Based on the reviews, we will not be able to accept this version of the manuscript, but we would be willing to review a much-revised version. We cannot, of course, promise publication at that time.

If you decide to revise the manuscript for further consideration at PLOS Genetics, please aim to resubmit within the next 60 days, unless it will take extra time to address the concerns of the reviewers, in which case we would appreciate an expected resubmission date by email to plosgenetics@plos.org.

[LINK]

We are sorry that we cannot be more positive about your manuscript at this stage. Please do not hesitate to contact us if you have any concerns or questions.

Yours sincerely,

Bruce Walsh

Guest Editor

PLOS Genetics

Gregory P.Copenhaver

Editor-in-Chief

PLOS Genetics

This is an important and interesting paper, suitable for a wide audience, and hence appropriate for PLoS Genetics. None of the three reviewers had any serious technical concerns (but see below for some suggestions), but they did express concerns in terms of the readability of the current ms for a general audience. Hence the recommendation of a major revision. A couple of suggestions beyond those offered by the reviewers. First, the quadratic components under inbreeding are a dark art for even most quantitative geneticists, and the notation varies widely even among its practitioners (W&L Table 11.2). That can make discussions hard to follow. Might I suggest that instead of D1 and D2, you use sigma_{ADI} and sigma_{DI} which have a bit more widespread (and also suggest what is being measured), and then define these (W&L Table 1), e.g., D1 = sigma(alpha_ki, delta_kii), the association between the additive effect of allele i (at locus k) and its dominance deviation (ii). While you do clearly say this, its a bit hard to follow unless you already know this, whereas I find the equation definition is usually clearer (and less ambiguous). Reviewer 3 notes that there are a lot of things going on in the ms, and to a reader outside the field, their interconnections may not be entirely clear, making the discussion a bit hard to follow (although the authors do nicely connect the dots at the end). In parts, it does feels a bit like a six pack of beer without the plastic rings that holds all the cans together. I think this can be a bit more clearly addressed with a short paragraph up front of what the paper will be looking for, and how these connect with each other (e.., variance components, rare QTLs, etc.). Another presentation suggestion is for the rare haplotype analysis (essentially a single allele test) to display the model in a slightly different from, y_ij = <cofactors> + x_{haij}*alpha + e_{ij}, where (new line) <cofactors> = all of the other stuff. The critical term can get buried in all of the model effects, whereas if they are clearly separated, the model becomes a bit more straightforward to an average reader.

Possible other things to consider

Some connection with heterosis would be interesting. It can be the flip side of inbreeding depression, or something entirely different. In particular I think a little more discussion on the role of loci around centromeres generating pseudo-overdominace might also be in order (a comment also suggested by reviewer #2)

Reviewer's Responses to Questions</cofactors></cofactors>

**Comments to the Authors:**

Reviewer #1: attachment.

Reviewer #2: Review is uploaded as an attachment.

Reviewer #3: The authors investigate genetic load and inbreeding depression in a pair of maize-teosinte populations evaluated for 18 traits using quantitative genetic theory and rare allele scans.

The research topic is very important and interesting for a broad readership, the data are unique, the analyses are statistically sound and innovative, the results are interesting.

The paper is very hard to follow and some paragraphs require deep knowledge of the subject to follow the line of argument.

The paper presents many correlations in the text as on page 9: “Narrow-sense heritability estimates based on additive genetic variances were highly correlated (r = 0.85 in teosinte and 0.83 maize) between the previous analysis of realized relationships and the current diallel model.

In many cases it is not clear where to find these results, what is expected, what is their relevance (two times same result?) and in general correlations do not say much as a high correlation can originate from all sorts of data structures.

Some results are confounded for example on page 9: “The cases of decreased

environmental variance may occur due to reduced scale associated with inbreeding, whereas the

increased environmental variance observed for maize vegetative and reproductive traits suggests that partially inbred maize plants have reduced capacity for homeostasis.”

How can these two factors be separated? The authors are careful with their conclusions but they still seem rather speculative.

The results are not sufficiently discussed and interpreted for example the results in Figure 3 and the paragraph on page 10 “The very high correlations between outbred and inbred progeny…at the same time.” Results are very important and have direct implications for the utilization of landraces or other genetic resources for breeding. The authors do not put this into context but leave it to the reader to make something of it.

The prediction of genetic variance among selfed families requires a more profound discussion. Variance components are generally not very precise, especially dominance variances. I suspect that “less than 90% expected variance” is not significantly different from 100%. In addition, averaged across traits it looks pretty close to 100%. A trait like tillering is probably not normally distributed and this can easily result in deviations from what is expected.

The scan for rare allele effects suffers from the same limitations as QTL mapping and GWAS. QTL are not reproducible across populations for many reasons (e.g. power etc.), effect estimates are biased especially for rare alleles. Effects need to be validated in an independent sample. The authors ignore these limitations and it is questionable if their conclusions are justified. Shouldn't effects in teosinte be predictive for maize if they share genomic regions? Wouldn't that be a nice validation?

My impression is that the paper would have profited a lot from a clear focus. At the moment it is more like an assembly of results that are not well connected by a common hypothesis or research question. The manuscript is also lacking interpretation of results beyond the analysis of inbreeding depression in a specific pair of populations. Grouping of the traits helped, but it would have been better to focus on a few traits. The data have been presented in other publications so no need to use all traits. The paper becomes rather unwieldy with many supporting figures and tables.

**Have all data underlying the figures and results presented in the manuscript been provided?**

Reviewer #1: None

Reviewer #2: Yes

Reviewer #3: Yes

PLOS authors have the option to publish the peer review history of their article (what does this mean?). If published, this will include your full peer review and any attached files.

Reviewer #1: **Yes: **John Kelly

Reviewer #2: **Yes: **Donald M. Waller

Reviewer #3: No

---

## [Decision Letter · Decision Letter 1]

3 Dec 2021

Dear Dr  Holland:

We are pleased to inform you that your manuscript entitled "Domestication Reshaped the Genetic Basis of Inbreeding Depression in a Maize Landrace Compared to its Wild Relative, Teosinte" has been editorially accepted for publication in PLOS Genetics. Congratulations!

Both reviewers who expressed concerns in the first round found the current ms appropriate for acceptance, but pointed out a few minor comments for you to consider as you prepare your final draft for the production team (the editorial team will not need to re-evaluate). Reviewer 1 (John Kelly, who signed is reviewed) detailed his comments (see below), while reviewer #2 made two minor comments to me that I wish to pass on:

1. l338-347 we know that the ratio of sca/gca has decreased over time with established heterotic groups and strong enrichment of favorable alleles in the respective pools, so this might be a bit of a contradiction to what they present here.

2. l489ff I think they did not use multiple testing when looking at the correlations, so I guess nothing is significant

Yours sincerely,

Bruce Walsh

Guest Editor

PLOS Genetics

Gregory P. Copenhaver

Editor-in-Chief

PLOS Genetics

Reviewer's Responses to Questions

**Comments to the Authors:**

Reviewer #1: The revised draft of this manuscript is greatly improved in terms of clarity of presentation. The authors have explained the analyses much more clearly in relation to previous publications on this dataset. They have also shifted the emphases in useful ways. I am a bit sorry to see the inbreeding variance component estimates diminished in the revised ms, but the authors give clear justification for this choice. I have only one further comment for consideration, and it has to do with the framing of the study. Lines 26-28: “The extent to which the genetic load of mutations contributing to inbreeding depression is due to rare large-effect variation versus potentially more common variants with very small individual effects is unknown…”

This sets up genetic variation as a 2x2 table (rows = rare versus intermediate allele frequency, columns = small effect versus large) where only two of the cells are filled (rare and large, intermediate and small). I think that it is likely that genetic load is also generated by the “missing” categories:

(1) In other systems, much inbreeding depression is generated by rare, small effect mutations at thousands of loci. I think the results of this study are consistent with a contribution of many rare small effect alleles given that the RAS loci do not fully explain inbreeding depression. This class of variants is interesting given that rare/small effect alleles are particularly hard to purge.

(2) In other systems, intermediate frequency polymorphisms of large effect (e.g. inversion polymorphisms) can generate substantial inbreeding depression, particularly when one homozygote is lame. In this experiment, I am not sure if this category of variant can be excluded. The QTL mapping did map some loci affecting the traits and a locus must have an appreciable affect to be detected by QTL mapping. The question then becomes – do all the QTLs with intermediate allele frequencies exhibit additive inheritance? If so, that would be good to include as reminder to the reader.

John Kelly

Lawrence, KS 11/15/2021

Reviewer #3: The authors made great efforts to increase the readability of the manuscript. The results of the study are now presented in a well structured way and can be much more appreciated by the reader. All my concerns have been addressed satisfactorily by the authors.

**Have all data underlying the figures and results presented in the manuscript been provided?**

Reviewer #1: None

Reviewer #3: Yes

PLOS authors have the option to publish the peer review history of their article (what does this mean?). If published, this will include your full peer review and any attached files.

Reviewer #1: **Yes: **John Kelly

Reviewer #3: No

**Data Deposition**

http://datadryad.org/submit?journalID=pgenetics&manu=PGENETICS-D-21-01141R1

**Press Queries**

---

## [Editor Report · Acceptance letter]

13 Dec 2021

PGENETICS-D-21-01141R1 

Domestication Reshaped the Genetic Basis of Inbreeding Depression in a Maize Landrace Compared to its Wild Relative, Teosinte 

Dear Dr Holland, 

We are pleased to inform you that your manuscript entitled "Domestication Reshaped the Genetic Basis of Inbreeding Depression in a Maize Landrace Compared to its Wild Relative, Teosinte" has been formally accepted for publication in PLOS Genetics! Your manuscript is now with our production department and you will be notified of the publication date in due course.

With kind regards,

Zsofia Freund

PLOS Genetics

On behalf of:
